

# Characterization of submicron aerosols influenced by biomass burning at a site in the Sichuan Basin, southwestern China

Wei Hu, Min Hu[*], Weiwei Hu, Hongya Niu, Jing Zheng, Yusheng Wu, Wentai Chen, Chen Chen, Lingyu Li, Min Shao, Shaodong Xie, Yuanhang Zhang

State Key Joint Laboratory of Environmental Simulation and Pollution Control, College of Environmental Sciences and Engineering, Peking University, Beijing 100871, China

*Correspondence to:* M. Hu (minhu@pku.edu.cn)

**Abstract.** Severe air pollution caused by large amount of pollutants and adverse synoptic processes appears often in Asia.

However, limited studies on aerosols have been conducted under high emission intensity, and unique geographical and meteorological conditions. In this study, an Aerodyne high resolution time-of-flight aerosol mass spectrometry (HR-ToF-AMS) and other state-of-the-art instruments were utilized at a suburban site, Ziyang, in the Sichuan Basin during December 2012 to January 2013. The chemical compositions of atmospheric submicron aerosols ($PM_1$) were determined, the sources of organic aerosols (OA) were apportioned, and the aerosol secondary formation and aging process were explored as well. Due to high

humidity and static air, $PM_1$ was maintained at a relatively stable level during the whole campaign, with the mean concentration of $59.7\pm24.1$ μg m$^{-3}$. OA was the most abundant component (36%) in $PM_1$, characterized by a relatively high oxidation state. Positive matrix factorization analysis was applied to the high resolution organic mass spectral matrix, which deconvolved OA mass spectra into four factors: low volatility (LV-OOA) and semi-volatile oxygenated OA (SV-OOA), biomass burning (BBOA) and hydrocarbon-like OA (HOA). OOA (sum of LV-OOA and SV-OOA) dominated OA as high as 71%. In total,

secondary inorganic and organic formation contributed 76% of $PM_1$. Secondary inorganic species correlated well with relative humidity (RH), indicating the humid air can favor the formation of secondary inorganic aerosols. With the increase of photochemical age, OA became more aged with higher oxidation state, and secondary organic aerosol formation contributed more significantly to OA. The slope of OOA against $O_x$ ($=O_3+NO_2$) steepened with the increase of RH, implying that besides

the photochemical transformation, the aqueous-phase oxidation was also an important pathway of the OOA formation. Primary emissions, especially biomass burning, resulted in high concentration and proportion of black carbon (BC) in $PM_1$. During the episode obviously influenced by primary emissions, the contributions of BBOA to OA (26%) and $PM_1$ (11%) were much higher than those (10-17%, 4-7%) in the clean and other polluted episodes, highlighting the significant influence of biomass

burning.

## 1 Introduction

With its dense population and rapid economic development in the past decades, the Sichuan Basin suffers from serious fine particle pollution and has become one of the most polluted regions in China. The basin, located in the southwest of China, is

one of the most populous regions in China and the world, with a population density of approximate 400 people per square kilometer. The two megalopolises, Chongqing and Chengdu, in the basin with the largest populations of about 30 and 14 million, respectively, have been seeing increases in industrial added values by an annual rate of over 10%. High emissions of gaseous and particulate pollutants, such as volatile organic compounds (VOCs), $SO_2$, organic carbon (OC), black carbon (BC) and fine particles ($PM_{2.5}$), are found in the Sichuan Basin over China (He, 2012). Adversely influenced by the particular

topographic condition, the Sichuan Basin is within the region of the lowest wind speed and relatively high humidity over China all year round (Chen and Xie, 2013; Yang et al., 2011). The highest annual mean aerosol optical depth (AOD) in the Sichuan Basin from 2000 to 2010 across China reflected the importance of large topography in aerosol accumulation (Luo et al., 2013). The unique geographical and meteorological conditions in the region favor the accumulation of local and regional atmospheric pollutants (Yang et al., 2011), making environmental threats in Sichuan more severe.

The Sichuan Basin has suffered from long-term poor visibility since the 1970s (Chen and Xie, 2012, 2013). The visibility degradation primarily results from anthropogenic pollutants and synoptic processes. Anthropogenic aerosols and moisture at the surface are the dominant determinants of the AOD, and the spatial distributions of both AOD and light extinction coefficient ($B_{ext}$) are strongly influenced by regional topography (Wang et al., 2013). Under the stable weather system, low visibility is strongly related to high relative humidity (RH) in Chengdu, and high RH, high pressure and low wind speed in Chongqing.





Since the 2000s, the air quality has been aggravated in the Sichuan Basin for more intense anthropogenic emissions (Chen and Xie, 2013). Severe visibility deterioration and frequent hazy days have become vital concerns in the Sichuan Basin.

In addition to the adverse topographical and meteorological conditions, variability of fine particle concentrations and physiochemical characteristics serves as another crucial factor in explaining the degradation of visibility in the Sichuan Basin. Yang et al. (2011) reported that during cold periods, high $PM_{2.5}$ levels at 129 and 156 µg m$^{-3}$ were observed in Chengdu and Chongqing, respectively. Organics and sulfate were identified as the main components (over 50%) in $PM_{2.5}$. High $SO_4^{2-}/NO_3^-$ ratio indicated that local and stationary sources were predominant in the region where high-sulfur coal fuels are consumed in large quantity; high $K^+$ concentration (more than 6 µg m$^{-3}$) in Chongqing during the winter suggested that biomass burning from residential heating also accounted for an important source of pollution (Yang et al., 2011; Cao et al., 2012). Wang et al. (2013) concluded that organics, ammonium bisulfate, ammonium nitrate, and moisture in fine particles contributed more than 86% to the $B_{ext}$ in Chengdu; biomass burning, coal combustion, vehicular and industrial emissions were the main contributors to both $PM_{2.5}$ and the light-scattering coefficient. Though several published papers focused on aerosol chemical and physical properties in the Sichuan Basin, highly time-resolved studies are rarely conducted. It is necessary to explore secondary formation in the influence of biomass burning by using high time resolution aerosol mass spectrum.

Organic aerosols (OA) are very significant components in fine particulate matter (Zhang et al., 2007). Several results on the compositions and sources of OA in the Sichuan Basin have also been reported. The concentration of OA in molecular level using GC/MS analysis was extremely high (9.7 µg m$^{-3}$ in winter) in Chongqing because of its active industrialization and urbanization. Anthropogenic sources, such as coal combustion, cooking and vehicle emissions, contributed to OA primarily. Levoglucosan occupied around 90% of total identified sugars in winter (700 ng m$^{-3}$) and summer (123 ng m$^{-3}$). The high levels of levoglucosan were most likely caused by biomass burning emissions via residential cooking and heating, especially in winter (Wang et al., 2006). Li et al. (2013a) found that about 15-21% of the OC could be apportioned to biogenic sources and processes, e.g., biomass burning, isoprene oxidation products and fungal spores, at a forest site and an urban site in Ya'an in the Sichuan Basin, during the summer of 2010. High organic and elemental carbon (OC and EC) concentrations and OC/EC ratio were observed in urban Chengdu, which revealed that the formation of secondary organic carbon (SOC) contributed to

OC as 55% based on the EC tracer method (Zhang et al., 2008). These results suggested the important contributions of biomass burning and other primary emissions, as well as secondary formation to OA in the Sichuan Basin.

Many studies refer to the particulate organic matter (OM), a product of atmospheric processes   including the oxidation of VOCs, shifting of chemical equilibrium, re-partitioning of semi-volatile species, adsorption/absorption through heterogeneous physical and chemical processes, and cloud physiochemical processes (Kroll et al., 2005; Hallquist, et al., 2009). Secondary organic aerosols (SOA) make up about 20-80% of OA in the atmosphere (Carlton et al., 2009), yet formation mechanisms of SOA remain essentially speculative, causing discrepancies between observed SOA and model simulations. What precursors and chemical mechanisms are important therefore maintains unclear (Hallquist, et al., 2009).

The employment of aerosol mass spectrometry (AMS, Aerodyne Research Inc., USA) not only can obtain high-resolution chemical composition of submicron aerosols, but also can allow source apportionment of primary organic aerosols (POA) and SOA (Ng et al., 2010). OA mass spectral matrix from Aerodyne AMS analyzed with a positive matrix factorization (PMF) technique (Paatero, 1997) resolved OA into several factors: oxygenated organic aerosol (OOA) factors described as low volatility and semi-volatile OOA (LV-OOA and SV-OOA), hydrocarbon-like (HOA), biomass burning (BBOA), cooking (COA) and coal combustion OA (CCOA), etc. (Jimenez et al., 2009; Ng et al., 2011; Hu et al., 2013, 2016). HOA, BBOA, COA and CCOA could be considered as POA; the OOA component has been shown to be a good surrogate SOA in many studies (Ng et al., 2010). Discrimination of different OA components can favor quantifying the primary and secondary contributions, and probing into secondary formation mechanisms and aging processes of OA (Ulbrich et al., 2009).

In this study, an Aerodyne high resolution time-of-flight AMS (HR-ToF-AMS) was deployed at a suburban site in Ziyang, downwind of Chengdu City in the Sichuan Basin, in the seriously polluted wintertime. We obtained highly time-resolved chemical compositions and size distributions of non-refractory submicron aerosols (NR-PM$_1$), apportioned the sources of OA, as well as investigated the secondary formation and aging process of OA under unique geographical and meteorological conditions. The results will give hints to the mechanism of haze formation and help control the serious air pollution in the Sichuan Basin.



## 2 Methodology

### 2.1 Sampling site

An intensive field campaign was carried out on the campus of a primary school in Ziyang (30.15° N, 104.64° E) during the wintertime from 3 December 2012 to 5 January 2013. The location of the observation site is shown in Fig. 1, and the topography of the Sichuan Basin is shown in Fig. S1 in Supplementary material. There were no obvious industrial sources around this site. Ziyang is a county-level city located in central Sichuan Basin, downwind of Chengdu Plain, and in between two megalopolises (90 km south of Chengdu and 260 km to the west of Chongqing). The observation site was selected and considered to be well representative to characterize air pollution in the Sichuan Basin. The 72-h backward trajectories of air parcels at Ziyang site at an altitude of 500 m during the campaign were calculated by NOAA's HYSPLIT4.9 model (www.arl.noaa.gov/hysplit.html), starting a new trajectory every 6 hours, and the result of clustering is shown in Fig. 1. The stagnant air prevailed in the one-month campaign due to the basin terrain. The only interruption of the atmospheric isolation was an invasion of long-distance transported air mass from northwest China accompanied with strong wind on 29 December. Therefore, the atmospheric processes are dominated by the isolated meteorology of the basin.

The HR-ToF-AMS was deployed along with other relevant measurement instruments to characterize chemical compositions of atmospheric submicron aerosols and evaluate the aerosol secondary formation and aging process. This is the first application of the HR-ToF-AMS in the Sichuan Basin. The collocated measurement instruments included a multi-angle absorption photometer (MAAP, Thermo Fisher Scientific Inc.) for BC, online gas chromatography-mass spectrometry/flame ionization detector (GC-MS/FID, TH-017, Wuhan-Tianhong Instrument Co.; Li et al., 2014) and proton transfer reaction-mass spectrometer (PTR-MS, Ionicon Analytik GmbH) for VOCs and an ambient air quality monitoring system for meteorological parameters and gaseous pollutant concentrations, etc. All instruments were placed in two containers settled on the open playground on the campus. The instrument setting, operation and data processing were carried out as described in Hu et al. (2013). Ambient particles were also collected on the copper mesh at the site in some days during the campaign. Particles were photographed and investigated using the transmission electron microscope coupled with an energy dispersive X-ray spectrometer (TEM-EDX, Tecnai G2 T20, FEI Corp.) in the Electron Microscopy Laboratory of Peking University.



## 2.2 HR-ToF-AMS operation and data processing

The HR-ToF-AMS measures the mass concentrations and size distributions of non-refractory species in submicron aerosols, including organics, sulfate, nitrate, ammonium and chloride (DeCarlo et al., 2006; Hu et al., 2013).

A PM$_{2.5}$ impactor inlet was set on the roof of the container to remove coarse particles. Airstream was introduced in through a copper tube at a flow rate of 8 L min$^{-1}$ and subsequently sampled into the HR-ToF-AMS at a flow rate of 0.09 L min$^{-1}$, isokinetically from the center of the copper tube. Before entering the instrument, the airstream was dried by a Nafion drier (Perma pure, Inc.), and kept RH below 30%. In order to make a better mass closure of refractory species, a MAAP mentioned above coupled with PM$_{2.5}$ cyclone was used for simultaneous measurement of BC with a 5-minute time resolution.

The HR-ToF-AMS operated in a cycle of 5 minutes during the campaign. Under the V-mode, it functioned on mass spectrum (MS) mode for 1 min to obtain the mass concentrations of the non-refractory species, and on separate PToF (particle time-of-flight) mode for 1.5 min to determine size distributions of species. Under the W-mode, only high resolution mass spectral data (HR-MS) was obtained for 2.5 min. The ionization efficiency (IE), sampling flow and particle sizing of HR-ToF-AMS were calibrated following the standard protocols (Drewnick et al., 2005). The calibrations of IE and the particle sizing used size-selected pure ammonium nitrate particles with nominal diameters of 400 nm and 60-650 nm, respectively. According to the definition of detection limits (DLs) of different species determined by AMS (Huang et al., 2011), the DLs (V-mode) of organics, sulfate, nitrate, ammonium, and chloride during the campaign were calculated to be 0.24, 0.07, 0.04, 0.05, and 0.01 μg m$^{-3}$, respectively.

V-Mode provides data with lower resolution while W-Mode produces data with higher one. V-mode data are used to generate unit mass resolution (UMR) spectra, from which mass concentrations and size distributions of species are determined (DeCarlo et al., 2006); W-mode data serve to separate ion fragments with the same nominal $m/z$ but different elemental compositions (Aiken et al., 2007). The standard AMS data analysis software packages (SQUIRREL version 1.57I and PIKA version 1.16H) downloaded from the ToF-AMS-Resources webpage (http://cires.colorado.edu/jimenez-group/ToFAMSResources) compiled and executed on Igor Pro 6.22A were used to generate UMR- and HR-MS from the V- and W-mode data, respectively. Here a CE factor of 0.5, which performed well in many previous field studies (Aiken et al., 2009), was used to calculate mass

concentrations. The default relative ionization efficiency (RIE) values (Jimenez et al., 2003) were applied in this study, except for ammonium for which RIE=4.04 was used assuming $PM_1$ was neutral. The reported O/C and H/C ratios of OA in previous studies are mostly biased low. In this study, the "Improved-Ambient" correction (Canagaratna et al., 2015) was applied to calculate the O/C and H/C ratios of OA. The "Improved-Ambient" corrected results of elemental ratios reported in previous studies are from Canagaratna et al. (2015) and Chen et al. (2015).

The technical details on Aerodyne HR-ToF-AMS data process, as well as the implementation and validation of the PMF results, could be seen in previous papers (e.g., Huang et al., 2010; Hu et al., 2013). The HR-MS (*m/z* 12-255) was analyzed by PMF model to identify major OA components, which can much better separate different OA components than UMR spectra (Aiken et al., 2009; DeCarlo et al., 2010). Elemental analysis of the OA components identified by PMF was carried out with the methods based on HR-MS as described previously (Aiken et al., 2007; Canagaratna et al., 2015). Besides evaluating the reliability and stability of the outcome of PMF model through several parameters (Sect. S3 in Supplementary material), the optimum solution can also be defined via comparing the output mass spectra with those of the known sources, comparing the time series of factors with external tracers, and analyzing the diurnal patterns of different factors, etc. (Zhang et al., 2011). The uncentered correlation (UC) coefficient of MS, i.e., the cosine of the angle between a pair of MS as vectors, was also used as a qualitative metric to support factor identification (Ulbrich et al., 2009). The UC coefficients between the MS of OA factors resolved in this study and the single or average MS of OA factors reported in previous studies are listed in Table S3. Based on all the tests, the four factors, $f_{Peak}$=0 and seed=0 solution was chosen as the optimal solution for this analysis.

## 3 Results and discussion

### 3.1 Chemical compositions and size distribution of $PM_1$

#### 3.1.1 Variations of chemical species

The time series of main chemical compositions of submicron aerosols and meteorological parameters during the observation period are shown in Fig. 2. Throughout the observation period, the average $PM_1$ mass concentration (sum of NR-$PM_1$ measured by AMS and BC by MAAP) was 59.7±24.1 μg m$^{-3}$. The lowest and highest $PM_1$ concentrations were 3.0 μg m$^{-3}$ in 29



December 2012 and 172.5 µg m$^{-3}$ in 1 January 2013, respectively. Organics, accounting for about 36.0±6.1% of PM$_1$, were identified as the most abundant components, followed by sulfate (20.5±4.7%), nitrate (15.0±5.2%), ammonium (13.8±2.3%), BC (11.1±3.1%) and chloride (3.5±3.5%). Compared with the reported results in China (Table S1), the submicron aerosol pollution during this campaign was at a higher level. PM$_1$ mass concentration at Ziyang site was comparable to that of Beijing, but higher than at other urban (Shanghai and Shenzhen), coastal/background (Backgarden and Changdao Island), and suburban/rural sites (Jiaxing, Kaiping and Heshan). At almost all of these sites, OM dominated submicron aerosols (about 30-40%); while at Ziyang site, the concentration of BC (6.5 µg m$^{-3}$) was significantly higher among these sites, indicating strong local primary emissions.

To illustrate the factors influencing air pollution, four episodes were selected according to different pollution levels (as Fig. 2c shown), including three pollution periods (Episode P1, P2 and P3) and a clean day (Episode C). The average PM$_1$ mass concentrations during Episode P1 and P2 were as high as 91.6 and 71.8 µg m$^{-3}$, respectively. During the whole campaign, calm occurred frequently (Fig. 2a). Only Episode C (29 December) is a perfectly sunny and clean day, due to the diffusion effect of strong wind (Fig. 2a). The PM$_1$ concentration plummeted to the lowest level, 7.6 µg m$^{-3}$ on average. Yet PM$_1$ concentration boosted rapidly in the following Episode P3 (30 December 2012 to 3 January 2013) due to apparent primary emissions. The concentrations of organics, sulfate, BC and chloride rose to different extents (Table S1), and the average PM$_1$ mass concentration was 56.9 µg m$^{-3}$. The local emissions may result from smoking bacon with biomass burning, a traditional and common method of preserving pork and sausages in the Sichuan Basin in winter.

The concentrations of secondary inorganic species (SNA, an acronym for sulfate, nitrate and ammonium) step-wisely increased during Episode P1 and P2, especially for nitrate, with much higher average concentrations (Table S1). The concentrations of SNA correlated well with RH (Pearson correlation coefficient r=0.536, 0.415 and 0.555, p<0.01), suggesting this was probably induced by the more effective secondary transformation in the humid air. In Episode C, as the strong wind from northwestern China caused atmospheric pollutant diffusion, the RH decreased to a minimum (Fig. 2b). Each chemical species in submicron aerosols was at the lowest level, which can be considered as the background concentration. The components of organics, sulfate, nitrate, ammonium, BC and chloride accounted for 38.1%, 27.5%, 6.2%, 13.8%, 10.9% and 3.4% of PM$_1$, respectively.

In Episode P3, the proportions of organics, BC and chloride accounted for $PM_1$ increased due to the strong primary emissions; hence those of SNA decreased. These results reflected the significant impact of meteorology conditions and emission sources on air pollution level.

Since the secondary compositions were dominant in $PM_1$, variation of $PM_1$ depended on secondary formation processes and removal processes, i.e., strong wind and wet deposition. During the campaign, six short-term precipitation events, the main removal approach in Ziyang, were observed, which could eliminate the heavy PM pollution partly (Fig. 2c). However, the humid air caused by the precipitation may favor the aqueous-phase secondary formation and hygroscopic growth of SNA in turn. The probability density of $PM_1$ mass concentration during the campaign followed normal distribution approximately (as the white curve shown in Fig. 3a), and concentrated in a mono-mode between 30 μg m$^{-3}$ and 80 μg m$^{-3}$, which was mainly caused by the adverse geological and meteorological conditions. The probability distribution of $PM_1$ mass concentration in Ziyang was similar to that in summer in Beijing, where the stagnant air mass prevented diffusion of pollutants, and high humidity and temperature favored secondary formation (Huang et al., 2010). It contrasted the distribution patterns observed at Changdao Island (Hu et al., 2013) and in Beijing (23 January to 2 March 2013; Fig. 3d) in cold period, with broader ranges of $PM_1$ mass concentration. During these two campaigns, clean days appeared more frequently due to the intruding clean air mass carried by strong wind. The probabilities of $PM_1$ mass concentration lower than 35 μg m$^{-3}$ were over 40%, while that was only about 10% in Ziyang. These results suggested the relatively stable state of submicron aerosol pollution at Ziyang site.

The proportion variations of different chemical species with the increase of $PM_1$ mass concentration are shown in Fig. 3a. The relative contributions of inorganic and organic varied insignificantly, with $PM_1$ mass concentration ranging from 35 μg m$^{-3}$ to 120 μg m$^{-3}$. When $PM_1$ mass concentration was below 35 μg m$^{-3}$, the fraction of organics increased slightly to 40% or above. High fractions of organics, chloride and BC caused by strong primary emissions were found as $PM_1$ was above 140 μg m$^{-3}$.

### 3.1.2 Size distribution of chemical species

The average vacuum aerodynamic ($d_{va}$) size distributions of the non-refractory species in submicron aerosols are shown in Fig. 3b. Organics and secondary inorganics featured similar size distribution patterns and existed primarily in accumulation mode

with peaks around 600-800 nm, implying that the particles were internally mixed. It was consistent with the morphology and mixing state of single particles, mostly spherical and in internal mixing state (Fig. 4a-d). The peak sizes of all species were larger than those in winter in Beijing (Fig. 3e), Mexico City and Changdao Island (Aiken et al., 2009; Hu et al., 2013), indicating that the aerosols at Ziyang site may be more aged than in other areas. Organics and chloride exhibited broader size distributions than SNA, with obvious mass enhancement at small sizes (100-500 nm), indicating contributions of primary emissions like biomass burning and coal combustion (Huang et al., 2010; Hu et al., 2013). With the increasing of particle size ($d_{va}$>200 nm), the proportion of organics contributed to submicron aerosols decreased slightly, while those of sulfate, nitrate and ammonium increased gradually (Fig. 3c), suggesting that SNA were the main contributors as the particles grew up in Ziyang. In contrast, sulfate made a more significant contribution to particle growth in Beijing winter (Fig. 3f).

## 3.2 Investigating OA sources with PMF

By conducting PMF analysis on the high mass resolution OA spectral matrix, four factors of OA were resolved, i.e., LV-OOA, SV-OOA, HOA and BBOA, with distinct mass spectral profiles (Fig. 5) and temporal variations (Fig. 6). They accounted for 34.7%, 36.5%, 14.9% and 13.9% of total OA mass, respectively, as shown in Fig. 8a. The former two factors are good surrogates of aged and fresh SOA, while the latter two are classified into POA, respectively (Jimenez et al., 2009). SOA (OOA) dominated in OA as much as 71%, approximate to or higher than the reported results in China (Table S1), implying the high oxidation state of OA in Ziyang. In total, secondary formation (SOA+SNA) contributed to PM$_1$ as high as 76%, in accordance with the results of single particle analysis (See Sect. S5). Further strengthening of gaseous precursor's control, therefore, should be pursued.

### 3.2.1 Hydrocarbon-like OA (HOA)

The average mass spectrum of HOA (Fig. 5) is similar to previously reported reference spectra of HOA (Aiken et al., 2009; Huang et al., 2010, 2011). In this spectrum, alkyl fragments are dominated, especially the saturated alkyl fragments ($C_nH_{2n+1}^+$) and the alkenyl fragments ($C_nH_{2n-1}^+$). The OM/OC and O/C ratios of HOA factor were about 1.31 and 0.10, respectively, which were approximate to those previous results (OM/OC: 1.34-1.43, O/C: 0.11-0.20) in other areas in China (Canagaratna



et al., 2015; He et al., 2011; Huang et al., 2012, 2013; Gong et al., 2012). Factor analysis (e.g., PMF analysis) suffers a limitation, as it is incapable of separating independent sources completely, and the resolved factor may be a mixture of various sources. The abundant characteristic fragments for COA (*m/z* 55, 57, etc.) and CCOA (*m/z* 67, 69, 91, etc.) can be seen in the mass spectrum of HOA factor (He et al., 2010; Hu et al., 2013). The scatter plot between $f_{55}$ vs. $f_{57}$ in Ziyang is shown in Fig.

7. HOA resolved in this study had high m/z 55 vs. m/z 57 ratio of 1.71, which was between the ratios (2.2-2.8) in COA and those (0.9-1.1) in other non-cooking POA components (Mohr et al., 2012). The ratios of $C_3H_3O^+/C_3H_5O^+$ and $C_4H_7^+/C_4H_9^+$ in HOA resolved in this study were 1.5 and 1.7, respectively, which were also in the ranges (1-2 and 1-2.5) between HOA and COA summarized by Mohr et al. (2012). HOA was well correlated (r=0.6-0.7, p<0.01) with primary source tracers (e.g., chloride, $NO_x$, BC, acetonitrile, and acetaldehyde), as shown in Table S4. Among them, chloride and acetonitrile are the tracers

of coal combustion and biomass burning emissions, respectively. The MS of HOA correlated well with the average MS of HOA factor reported in previous studies, as well as that of COA, BBOA and vehicle emitted OA (Vehicle-OA) factors (Table S3). Thus, it was likely that the HOA factor was a mixture of COA and other primary organic aerosols. The diurnal cycle of HOA presented a peak at around 10:00, which may be related to the living habit of local residents. In rural and suburban areas in southwestern China, residents usually have two meals a day, especially in winter due to short daytime and less labor, so the

influence of cooking emissions was weaker at noon.

**3.2.2 Biomass burning OA (BBOA)**

Biomass burning via cooking and house heating plays an important role in aerosol pollution in the wintertime in southwestern China (Wang et al., 2006; Cao et al., 2012). Levoglucosan is an important tracer of biomass burning aerosols, the fragment of which, $C_2H_4O_2^+$, contributed to *m/z* 60 is regarded as a tracer ion of BBOA. The highest abundance of *m/z* 60 (~1.3%) is a

prominent characteristic in the MS of BBOA factor (Fig. 5), which is much higher than that (0.3%) in plumes with negligible biomass burning influence (Cubison et al., 2011). The MS of BBOA resolved in this study showed good correlation with the average MS of BBOA in previous studies (Table S3). In addition, BBOA tracked well with $C_2H_4O_2^+$ (r=0.85), further certifying the rationality of the resolved BBOA factor (Table S4). Compared with the O/C ratio of HOA (0.10), that (0.32) of BBOA was higher and in the range of 0.25-0.55 reported previously (Canagaratna et al., 2015). There was a phenomenon of burning straw

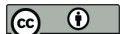

and wood randomly, especially which is an effective energy for cooking and heating in winter in Southern China (Song et al., 2009). High BBOA contribution (14±7%) to OA was consistent with significant biomass burning contribution (9-37%) to atmospheric VOC species during the same campaign (Li et al., 2014). Soot aggregates and sulfur-, chlorine- and potassium-containing particles were observed using TEM-EDX (Fig. 4e), also implying that biomass burning was a major contributor to

aerosol pollution (See Sect. S5). BBOA was probably emitted from residential houses, makeshift stoves built by migrant workers nearby, and waste incineration observationally.

BBOA displayed a very similar diurnal pattern with HOA and BC, with lower concentrations during the daytime and higher ones in the morning and nighttime (Fig. 8b and Fig. S4), indicating that they may be emitted by similar processes, such as residential emissions via cooking, heating and smoking bacon. The previous emission inventory (Guo et al., 2015) showed

that residential sources of OA were significant in the Sichuan Basin during the winter 2010 (Fig. 1). All the patterns were likely corresponding to the living habits of local residents as well as the diurnal variation of atmospheric boundary layer. The time series of BBOA tracked well (r=0.67) with another tracer of biomass burning events, acetonitrile (Fig. 6). BBOA also presented good correlations with BC and acetaldehyde (Table S4), which were mainly emitted from primary sources. In Episode P3, the contributions of BBOA to total OA (26%) and $PM_1$ (11%) were much higher than those (10-17%, 4-7%) in

other episodes defined above (Table S1), indicating far stronger biomass burning during Episode P3.

### 3.2.3 Semi-volatile and low-volatility oxygenated OA (SV-OOA and LV-OOA)

In a large number of studies, OOA has been widely investigated (Zhang et al., 2005; Aiken et al., 2009; Ng et al., 2011; Hu et al., 2016), which is considered a good alternative to SOA as the influence of biomass burning is negligible. As Fig. 5 shown, the identified mass spectra of LV-OOA and SV-OOA are both characterized by the oxygenated fragments ($C_xH_yO_z^+$), mainly

from carboxylic acid and aldehyde, especially $CO_2^+$ ($m/z$ 44) and $C_2H_3O^+$ ($m/z$ 43). The abundance of $C_xH_yO_z^+$ in LV-OOA was higher than that of SV-OOA. In this study, the abundances of $CO_2^+$ in LV-OOA and SV-OOA were 18% and 14%, respectively. LV-OOA with higher OM/OC and O/C ratios (2.52 and 1.02) was more oxygenated and aged than SV-OOA with lower ratios (2.12 and 0.73). The O/C ratios of LV-OOA and SV-OOA at Ziyang site were quite higher than the average O/C ratios for LV-OOA (0.84) and SV-OOA (0.53) summarized by Canagaratna et al. (2015), indicating OA was highly oxygenated

in Ziyang. Besides, the effect of biomass burning cannot be completely ruled out, which may also result in higher O/C ratio of SV-OOA.

Generally, OOA tracks well with SNA. LV-OOA correlates better with sulfate, and SV-OOA correlates better with nitrate, for their common volatility. In this study, LV-OOA correlated well with SNA (r=0.66-0.68), while the correlation between SV-OOA and SNA were slightly weaker (Table S4). The time series of SV-OOA roughly trended well towards that of nitrate except two intervals, 7-10 and 24-26 December. The total OOA (LV-OOA and SV-OOA) and secondary inorganic species (sulfate and nitrate) displayed good correlation (r=0.88), in accordance with the dominant secondary origin of OOA. LV-OOA also showed a similar trend to BC (r=0.75), maybe because BC was difficult to diffuse and mixed well in the static air.

The diurnal variation of LV-OOA showed a valley in the early morning (Fig. 8b, c), and increased from 9:00 to 14:00, indicating a significant secondary formation in the daytime. It displayed another peak in the evening, which may be influenced by the low-volatility particle-phase compounds from biomass burning (Murphy et al., 2014). The SV-OOA concentrations showed a small peak in the afternoon for more efficient photochemical reactions. The fractions of LV-OOA in OA exhibited a relatively stable diurnal pattern, implying LV-OOA was of regional characteristics. The fractions of SV-OOA in total OA varied a little diurnally and only increased slightly to 30% in the afternoon (13:00-18:00). Conversely, the fractions of HOA and BBOA decreased in the corresponding interval.

Depending on the pollution severity, the contributions of OOA components were distinct. In Episode P1 and P2, OOA was prominent (66% and 76%) in OA, and especially LV-OOA dominated in total OOA as 78% and 53%, respectively. In Episode C, SV-OOA predominated in total OA and OOA as high as 61% and 98%, respectively. While, in Episode P3, POA (sum of HOA and BBOA) and OOA almost contributed equivalently on average, and the contribution (56%) of SV-OOA to OOA was higher than that of LV-OOA (Table S1).

The proportion of each OA component in total OA as a function of OA concentration and the probability density distribution of OA concentrations are shown in Fig. 8d. OA concentrations were approximately skewed-normally distributed, and mainly in the range of 10-40 μg m$^{-3}$. In this range, SOA especially LV-OOA dominated the increase of OA concentration, contributed up to over 80% of OA. As the OA concentration was below 10 μg m$^{-3}$, SV-OOA transformed from gaseous precursors

contributed to OA predominantly. When the OA concentration was greater than 50 µg m$^{-3}$, OA was mainly from strong primary emissions, such as biomass burning. The proportion of LV-OOA and SV-OOA in OA decreased, while the fraction of POA increased dramatically. Specifically, corresponding to strong primary emissions during Episode P3, OA was mainly composed of POA and SV-OOA (as the spikes shown in Fig. 8d) due to the clean background atmosphere. Due to the highly oxidized and aged state of SV-OOA in Ziyang, the MS of SV-OOA was quite similar to that of LV-OOA (Table S3). However, they contributed discriminatively at different OA concentrations (Fig. 8d), which should be consistent with the trend of OOA oxidation.

### 3.3 Secondary formation and aging process of OA

The aging process plays an important role in the life cycle of atmospheric aerosols. The chemical composition, hygroscopicity and solubility of atmospheric aerosols are varied due to the aging process, which not only influences their optical properties and capability to form clouds, but also changes their impacts on environment, climate and human health. The aging process of OA can be characterized by some metrics and tools, including C/H/O atomic ratios, van Krevelen diagram (VK diagram), OM/OC ratio, OA/ΔCO, average carbon oxidation state ($\overline{OS_C}$), and the abundance of characteristic fragment ions ($f_{43}$ and $f_{44}$), etc., to provide the basis for model simulation of SOA formation (de Gouw and Jimenez, 2009; Kroll et al., 2011; Hu et al., 2013).

### 3.3.1 Elemental compositions of OA and VK diagram

The HR-MS obtained in this campaign was used to calculate the elemental compositions and OM/OC ratio of organic aerosols. During the overall observation period, on average, C, H, O and N contributed 49.5%, 6.4%, 42.9% and 1.2% to the total organic mass, respectively. As for atomic number ratios, average elemental ratios of O/C, H/C, and N/C were 0.65±0.11, 1.56±0.06, and 0.02±0.00, respectively. The average O/C and H/C ratios were close to O/C ratios (0.6) for downwind locations, and H/C ratios (1.5) for remote/rural locations (Chen et al., 2015). The average OM/OC (2.02±0.14, in the range of 1.37-2.35) and O/C ratios were higher than those in urban and suburban/rural areas in China (Table S1), indicating that OA was highly oxidized in Ziyang. Due to primary emissions from 29 December 2012 to 2 January 2013 (Episode P3), the O/C ratio declined apparently;

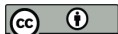



while the H/C ratio varied oppositely. In Episode C and P3, the O/C ratios (0.46, 0.52) were much lower than those (0.65, 0.69) during Episode P1 and P2 (Table S1). As local primary emissions contributing to OA substantially (the uppermost data points in the plot), the O/C ratio decreased to the lowest level, and the H/C ratio reached up to about 1.9.

The VK diagram, displaying the variation of O/C versus H/C, can be used as a tool to probe bulk oxidation reaction mechanisms for organic aerosols (Hu et al., 2013). The slope and intercept of VK diagram for OA at Ziyang site was −0.44 ($r^2$=0.70) and 1.84, respectively (Fig. 9a). The slope was shallower than those (−1.0 to −0.7, −0.58±0.04 for mean fit) observed across the world, which may be more consistent with chemical aging, and the intercept fell into the range (1.8-2.2) for remote/rural sites (Chen et al., 2015). The slope was close to −0.5, which suggests net changes equivalent to the addition of acid groups with fragmentation and/or both acid and alcohol/peroxide functional groups without fragmentation (Ng et al., 2011). During the Episode P1, P2, C and P3, the slopes of VK diagram were −0.46, −0.57, −0.84 and −0.52, respectively. The intercepts of the fitting lines were about 1.87-1.95. This result indicated that carboxylic acid functionalization with fragmentation was dominated during the pollution episodes, while carboxylic acid functionalization without fragmentation or addition of an alcohol and carbonyl group on different carbons was more active during the clean episode.

The photochemical age metric, −log ($NO_x/NO_y$), was used to investigate the relationship between OA oxidation and the photochemical aging. The ratio is higher in the more aged plume (Decarlo et al., 2008, 2010). When the metric −log ($NO_x/NO_y$) <0.1, it is considered to be fresh plume (Liang et al., 1998). Coloring the scattering data points in VK diagram (Fig. 9a) with −log ($NO_x/NO_y$), there was a clear trend (from the upper left to the lower right) that the aged plumes agreed with higher OA oxidation levels (higher O/C ratios), except for the data in the clean day (7:00-18:00 29 December; lower middle). The plume was probably well aged photochemically due to strong solar radiation, and the SOA was predominated by freshly formed SV-OOA (Fig. 8d). The OA factors resolved by AMS-PMF analysis are also marked in Fig. 9a. In the order from POA (HOA and BBOA) to SOA (SV-OOA and LV-OOA), the OA factors evolved along with the direction to a higher oxidation state, which was consistent with the oxidation characteristics of the factors (Ng et al., 2011), although SV-OOA evolved along a line with a smooth slope to LV-OOA.



### 3.3.2 Triangle plot ($f_{44}$ vs. $f_{43}$ and $f_{44}$ vs. $f_{60}$)

OA evolution can also be characterized in terms of the varying abundances of the two most dominant oxygen-containing ions in the OOA spectra, $m/z$ 43 and $m/z$ 44 (mostly $CO_2^+$ in ambient data). Since $m/z$ 44 is found to be proportional to the acid species, it seems that acid group formation plays a significant role in OOA aging process (Ng et al., 2011). The $m/z$ 43

fragments are mainly $C_2H_3O^+$, predominantly due to non-acid oxygenates, for the OOA fraction, and $C_3H_7^+$ for the HOA fraction. To avert the effects of atmospheric dispersion and dilution capability, $m/z$ 43 and $m/z$ 44 fractions in organic mass spectra ($f_{43}$ and $f_{44}$) were used to characterize the oxidation of OA.

The scatterplot of $f_{44}$ against $f_{43}$ is shown in Fig. 9b, and colored with $-\log (NO_x/NO_y)$. The $f_{44}$ ranged from 0.03 to 0.17, and the $f_{43}$ was in a narrow range of 0.05 to 0.09, which fitted the triangle space for OA components (Ng et al., 2011). With

increasing photochemical age ($-\log (NO_x/NO_y)$), $f_{44}$ increased and $f_{43}$ decreased, reflecting the photochemical aging of OA (Ng et al., 2010). The locations of OA factors are also marked in the plot. OA showed the evolution trends moving from the bottom (HOA and BBOA, $f_{44}<0.05$), to an intermediate location (SV-OOA), and to the top (LV-OOA) in the triangle. Data points gradually move upward from the lower half of the triangle with enhancement of OA oxidation in smoke chamber experiments and field observations (Ng et al., 2010). The location of SV-OOA on the upper half of the triangle is close to that of LV-OOA

(Fig. 9b), highlighting the high oxidation level of SV-OOA.

The scatterplot of $f_{44}$ against $f_{60}$ (Fig. 9c), colored with $-\log (NO_x/NO_y)$, was also applied here to facilitate understanding the secondary formation and transformation of primary BBOA in Ziyang. Cubison et al. (2011) reported that in the $f_{44}$ against $f_{60}$ space, data with negligible biomass burning influence were concentrated on the left side as a band shape ($f_{60}$=0.2-0.4%), while data from biomass burning appeared in the lower right part. Almost all data points fell into the left side of the conceptual space

for BBOA (Cubison et al., 2011), indicating the important contribution of biomass burning to OA during the whole observation period. HOA and SV-OOA resolved in this study were located out of the conceptual space for BBOA, while BBOA and LV-OOA were located in it. In addition, $m/z$ 60 accounted for 0.8%, 0.5%, 0.7%, and 1.3% of LV-OOA, SV-OOA, HOA, and BBOA, respectively. Higher abundance of $m/z$ 60 indicated that LV-OOA and HOA were probably associated with biomass burning processes and had been referred to as LV-bbSOA and bbPOA (bb, biomass burning) by Murphy et al. (2014). There

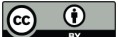


was no trend for the variation of $f_{60}$ of OA upon the increase of $-\log(NO_x/NO_y)$, suggesting that there was no dependence of the contribution of biomass burning to OA on photochemical aging.

### 3.3.3 Average carbon oxidation state of OA

The average oxidation state of the carbon ($\overline{OS_C} \approx 2 \times O/C - H/C$) is an ideal metric for the degree of oxidation of organic species in the atmosphere, and serves as a key quantity to describe organic mixtures that are as chemically complex as organic aerosols (Kroll et al., 2011). The $\overline{OS_C}$ in Ziyang was calculated as $-0.26\pm0.27$ on average, ranging from $-1.59$ to $0.36$. It was within the $\overline{OS_C}$ ranges in downwind and remote/rural environment (Chen et al., 2015), lower than those of ambient OA in the aged (Whistler Mountain) and coastal/background (Changdao Island) atmosphere, much higher than or comparable to those of other urban and suburban sites in China and laboratory-generated POA and SOA (Fig. 10 and Table S1), also implying OA was highly oxygenated.

A strong correlation between $\overline{OS_C}$ and $f_{44}$ (Fig. 9d) indicated that carboxylic groups can fragment into a large amount of $CO_2^+$ ions, and explain for the high $\overline{OS_C}$ observed. The fitted line for the overall data has a slope of 11.5 and an intercept of $-1.64$, suggesting that the $\overline{OS_C}$ of non-acid moieties in OA was $-1.64$. The $-\log(NO_x/NO_y)$ increased with increasing $\overline{OS_C}$ and $f_{44}$ (Fig. 9d), indicating that it is a good qualitative clock for photochemical age (Decarlo et al., 2008), except for the data of 29 December as mentioned above. The average $\overline{OS_C}$ of HOA, BBOA, SV-OOA and LV-OOA were $-1.61$, $-1.04$, $-0.19$ and $0.59$, respectively. The results were in the ranges of previously reported ones (Fig. 10), and in accordance with that $\overline{OS_C}$ must increase upon oxidation of OA and those non-acid oxygenated groups may undergo further oxidation during their atmospheric lifetime if conditions permit (Kroll et al., 2011). The $\overline{OS_C}$ in Episode C and P3 were decreased due to the freshly emitted organic aerosols; while in Episode P1 and P2, the $\overline{OS_C}$ increased to $-0.65$ and $-0.69$ (Fig. 10 and Table S1), indicating the OA during the hazy periods may contain relatively more abundant oxygenated groups other than the carboxylic group, such as carbonyl and hydroxyl groups (Li et al., 2013b).



### 3.3.4 Evolution of OA/ΔCO ratio with chemical conversions

In order to understand the effects of chemical conversions on the properties of organic aerosols in the atmosphere, it is necessary to avoid the influences of emissions and transport of OA by normalizing OA to a relatively inert combustion tracer over the time scales of interest, e.g., CO (de Gouw and Jimenez, 2009). The OA/ΔCO ratios are used to evaluate the secondary

formation of OA, where ΔCO indicates that the regional background concentration of CO (0.2 ppmv, the average concentration in Episode C) has been subtracted. OA/ΔCO ratio is lower for the primary emission plume from source region, and becomes higher after substantial SOA formation (DeCarlo et al., 2010). The average OA/ΔCO ratio in Ziyang during the campaign was 41.7±23.0 µg m$^{-3}$ ppmv$^{-1}$, which was lower than the average level (70±20 µg m$^{-3}$ ppmv$^{-1}$) around the world (de Gouw and Jimenez, 2009). In more polluted periods, SOA/ΔCO accounted for 70-80% of OA/ΔCO, implying higher contribution of

secondary formation. The influence of biomass burning emission can also cause high OA/ΔCO ratio, often similar to or even higher than SOA/ΔCO ratio from aged plumes (Cubison et al., 2011). In Episode P3, the OA/ΔCO reached the highest value as 209.2 µg m$^{-3}$ ppmv$^{-1}$. It was similar to the ever reported highest results 210 µg m$^{-3}$ ppmv$^{-1}$ in Mexico City when influenced by strong biomass burning emissions (Decarlo et al., 2010), indicating the important contribution of biomass burning to OA. Besides, SV-OOA/ΔCO increased from 7.7 (P1) and 16.1 (P2) to 19.8 µg m$^{-3}$ ppmv$^{-1}$ for SV-OOA can be quickly formed in

the plumes.

Photochemical age was calculated using ratios of m+p-xylene to ethylbenzene concentrations with an initial emission ratio of 2.2 ppbv ppbv$^{-1}$ (Fig. S5, Yuan et al., 2013). The average OH radical concentration applied here is 1.6×10$^6$ molecule cm$^{-3}$ in order to compare with other studies (DeCarlo et al., 2010; Hu et al., 2013). A detailed description of the determination of photochemical age can refer to Yuan et al. (2013). The variations of PMF resolved OA factors to ΔCO as a function of

photochemical age are shown in Fig. 11a. With the increase of photochemical age, POA components (BBOA and HOA) maintained at a stable level, implying the stable background concentration of POA in the daytime. However, roughly, SOA surrogate components (LV-OOA and SV-OOA) enhanced with the increase of photochemical age, which was consistent with the photochemical processing of OA. Specifically, the regression slopes of average LV-OOA/ΔCO and SV-OOA/ΔCO versus photochemical age in the range of 2.6-7.1 hours were 0.48 µg m$^{-3}$ ppmv$^{-1}$ h$^{-1}$ and 0.60 µg m$^{-3}$ ppmv$^{-1}$ h$^{-1}$ (Fig. S6), respectively,

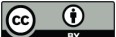



which may result from the efficient secondary formations from plenty of emitted POA, especially BBOA (Robinson et al., 2007). The increase of OA (1.2 μg m$^{-3}$ ppmv$^{-1}$ h$^{-1}$) was almost completely attributed to the contribution of SOA, which was approximate to that reported at Changdao Island (1.3 μg m$^{-3}$ ppmv$^{-1}$ h$^{-1}$) and lower than the ratios (2-5 ppmv$^{-1}$ h$^{-1}$) reported in Mexico City and the US (Hu et al., 2013). As the photochemical age was longer than 7 hours, the average OA/ΔCO ratio

decreased with photochemical age, caused by the decrease of LV-OOA/ΔCO ratio. Given the concentrations of OA components in aged air during this campaign, the evolution from POA, to SV-OOA, and to LV-OOA may be inhibited by lower concentration of POA (about one third of those in fresh air), resulting in lower LV-OOA concentrations, but relatively stable SV-OOA concentrations.

The average fractions of OA components in total OA at each bin versus the photochemical age are shown in Fig. 11b. SOA

dominated OA (56-84%) in both fresh and aged plumes, suggesting the high oxidation state of OA. When the photochemical age was nominally very short, POA (HOA+BBOA) accounted for 44% of total OA. While, due to the unique geographical and meteorological conditions in the basin terrain, it is reasonable that the aged OA is mixed with freshly emitted gaseous pollutants in the air, resulting in the substantial fraction of OOA at low apparent ages (Hu et al., 2013). The POA fraction in total OA decreased rapidly with the increasing of photochemical age. The percentage of SV-OOA (54%) was much higher

than that of LV-OOA (27%) in total OA in aged plumes, implying that the photochemical formation of SV-OOA was more efficiently than that of LV-OOA during this campaign.

The good correlations between OOA and O$_x$ (O$_x$=O$_3$+NO$_2$, surrogate of total oxidant) were considered to be useful for empirical predictions of SOA productions in previous studies in Mexico City and Houston (Herndon et al., 2008; Wood et al., 2010). However, OOA didn't correlated well with O$_x$ in Ziyang (Fig. S7), indicating the SOA formation mechanisms in Ziyang

differed greatly from those in Mexico City and Houston. The average RH in Ziyang during the campaign was 80±19% (12-100%), and reached saturation frequently (Table S1). Colored the scatter plot with RH, it can be found that the slope of OOA against O$_x$ steepened with the increasing RH (Fig. S7), indicating that both photochemical and aqueous-phase oxidation can dominate the secondary formation of OA in the atmosphere in Ziyang (Hu et al., 2016). In drier air (RH<40%), OOA formation

was dominated by photochemical processes, while the aqueous-phase oxidation became a more significant and efficient approach to OOA production in humid atmosphere (RH>40%) in Ziyang probably.

## 4 Conclusions

We investigated the chemical compositions of atmospheric submicron aerosols with a HR-ToF-AMS at a suburban site, Ziyang, located in the Sichuan Basin, China during the wintertime from December 2012 to January 2013. This study provided a special case of studying the characteristics and sources of aerosol pollution under the specific geographical and meteorological conditions in the basin terrain.

The mass concentrations of $PM_1$ maintained at a moderate level (59.7±24.1 μg m$^{-3}$) during the whole campaign. OA was the most abundant $PM_1$ component (36%). High OM/OC, O/C ratios and average carbon oxidation state indicated that organic aerosols were in a high oxidation state. Using AMS-PMF analysis, four OA fractions defined as LV-OOA (34.7%), SV-OOA (36.5%), HOA (14.9%) and BBOA (13.9%) were identified. Secondary formation contributed predominantly to OA (71%) and $PM_1$ (76%). Secondary inorganic species (SNA) contributed significantly to the heavy aerosol pollution, due to the more effective secondary formation and hygroscopic growth in the humid air. The OA factors evolved along with the direction to a higher oxidation state, from POA (HOA and BBOA) to SOA (SV-OOA and LV-OOA). With the increase of photochemical age, OA became more aged with higher oxidation state (higher O/C ratio, $f_{44}$, and $\overline{OS_C}$), and LV-OOA/ΔCO and SV-OOA/ΔCO also increased, implying photochemical processing contributed significantly to OA. The photochemical formation of SV-OOA was more efficient than that of LV-OOA during this campaign. The aqueous-phase oxidation can also contribute significantly to SOA production in humid atmosphere, with the OOA/$O_x$ ratio and RH increased simultaneously to some extent.

The concentration and proportion of BC in $PM_1$ at Ziyang site were at a much higher level among those reported results in China, indicating the severe primary emissions. During the episode obviously affected by primary emissions, the contributions of BBOA to OA and $PM_1$ were much higher than those in other polluted episodes, highlighting the important influence of biomass burning.



These results provide a better understanding of the role of primary emissions and secondary formation in submicron aerosol pollution in the Sichuan Basin. In the future, further work should be done to elucidate more details of the haze formation mechanisms, and to assess the effects of aerosol pollution in the Sichuan Basin.

**Acknowledgements**

5   This work was supported by the National Basic Research Program of China (2013CB228503), the Strategic Priority Research Program of the Chinese Academy of Sciences (XDB05010500) and the China Ministry of Environmental Protection's Special Funds for Scientific Research on Public Welfare (20130916). We would like to thank the observation team in Ziyang for their kind help, in particular for Sichuan Provincial Monitoring Center. We appreciate Dr. D. R. Worsnop from Aerodyne Research Inc., and Dr. S. Guo and Dr. J. Peng in Peking University (PKU) for their helpful comments, Mr. Z. Gong in PKU Shenzhen

10   for his guidance of data processing, and Prof. Q. Zhang in Tsinghua University for sharing the database of Multi-resolution Emission Inventory for China (MEIC). We thank Miss Y. Tian from Cornell University and Prof. J. Morrow from the Prefectural University of Kumamoto for their English polishing.



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





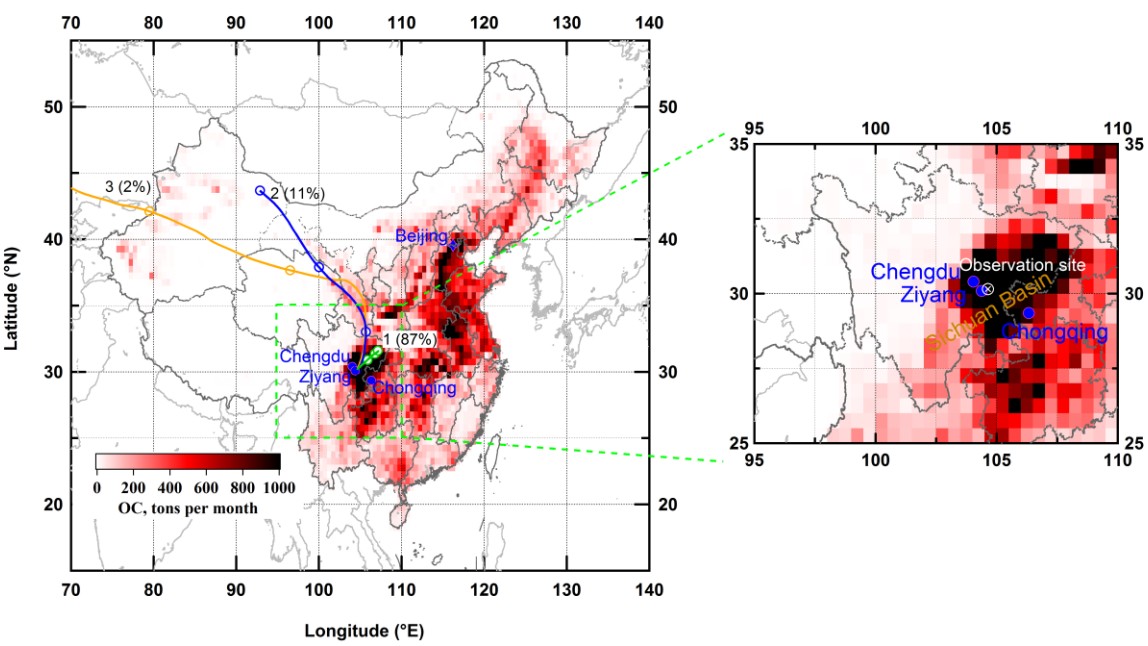

Figure 1. Location of the observation site in Ziyang in the Sichuan Basin. Back-trajectories of air masses at the site calculated by HYSPLIT model are illustrated as lines (circles marking 24-h intervals). The map of China is color-coded according to residential OC emissions in January 2010 modeled by Multi-resolution Emission Inventory for China (MEIC, http://www.meicmodel.org).



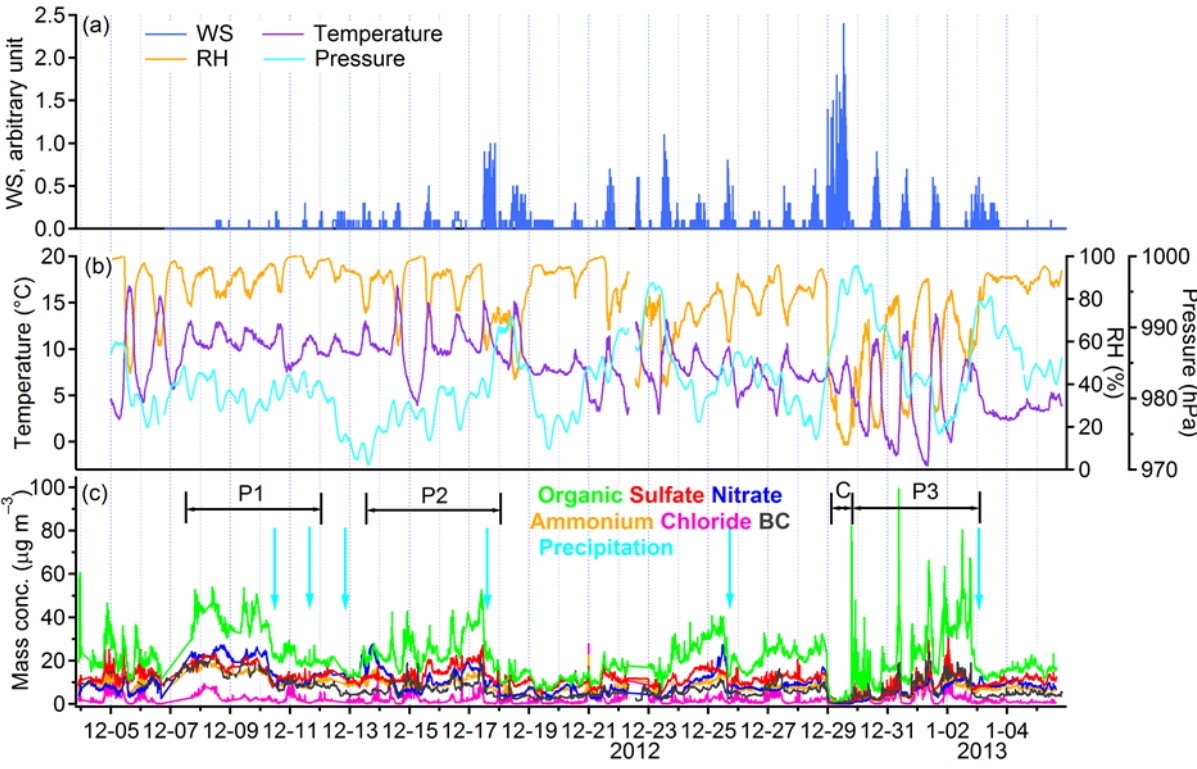

Figure 2. Time series of meteorological parameters and concentrations of chemical compositions in submicron aerosols during the campaign. (a) Wind speed (WS), relative values; (b) relative humidity (RH), temperature and atmospheric pressure; (c) concentrations of chemical compositions in submicron aerosols. Short-term precipitation events are marked by the light blue arrows.





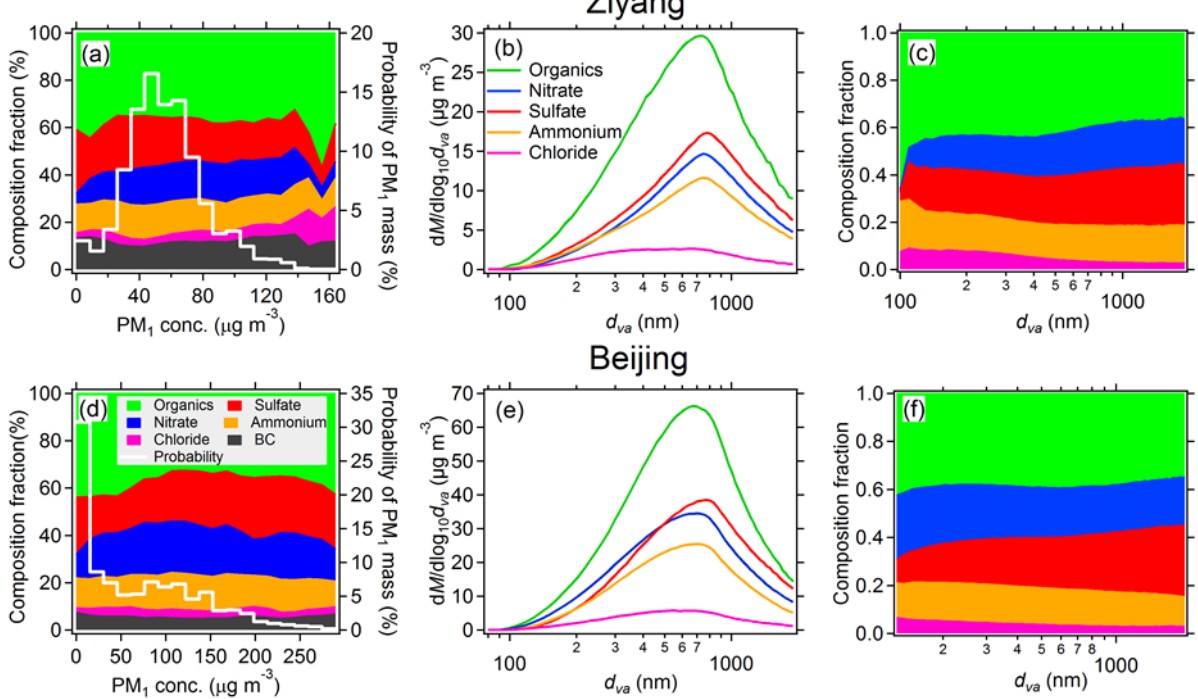

Figure 3. Comparison of the results between Ziyang site (upper) and an urban site in Beijing (lower) during the near wintertime.

(a, d) Fractions of main chemical components in $PM_1$ as a function of $PM_1$ mass concentrations (left) and the probability

density of $PM_1$ mass concentrations (right); (b, e) Average size-resolved mass concentration distributions of chemical species

5    in submicron aerosols; (c, f) Fractions of chemical species in total NR-$PM_1$ as a function of vacuum aerodynamic size ($d_{va}$).



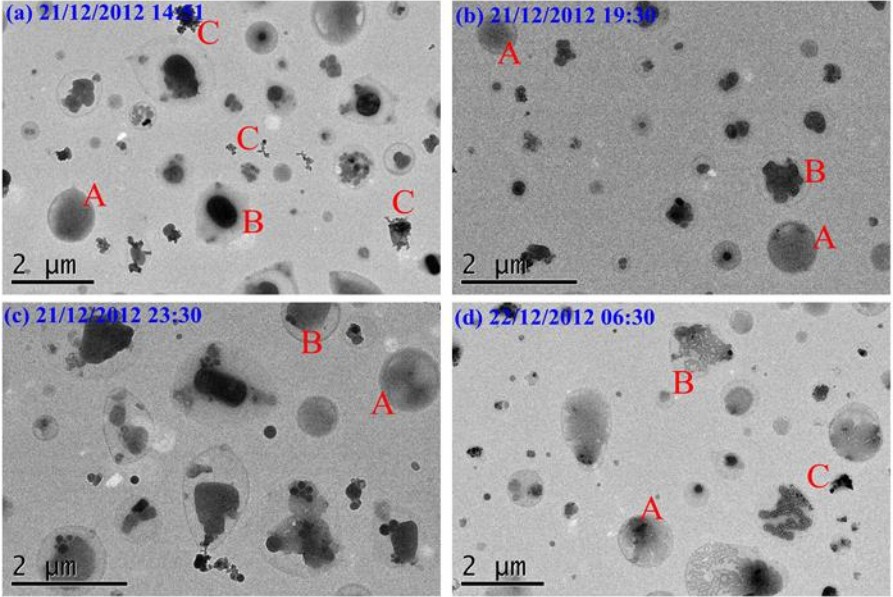

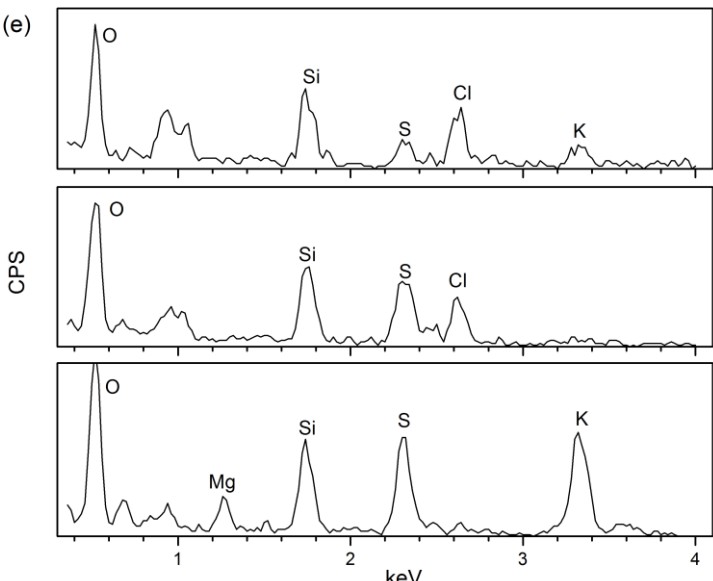

Figure 4. (a-d) Morphology and the mixing state of single particles in hazy days at Ziyang site. The predominant particles are spherical ones without or with coating (Type A and B), and in internal mixing state; some fresh and aged soot aggregates (Type C) were also observed. (e) Examples of elemental compositions in single particles collected at Ziyang site. CPS, counts per second.



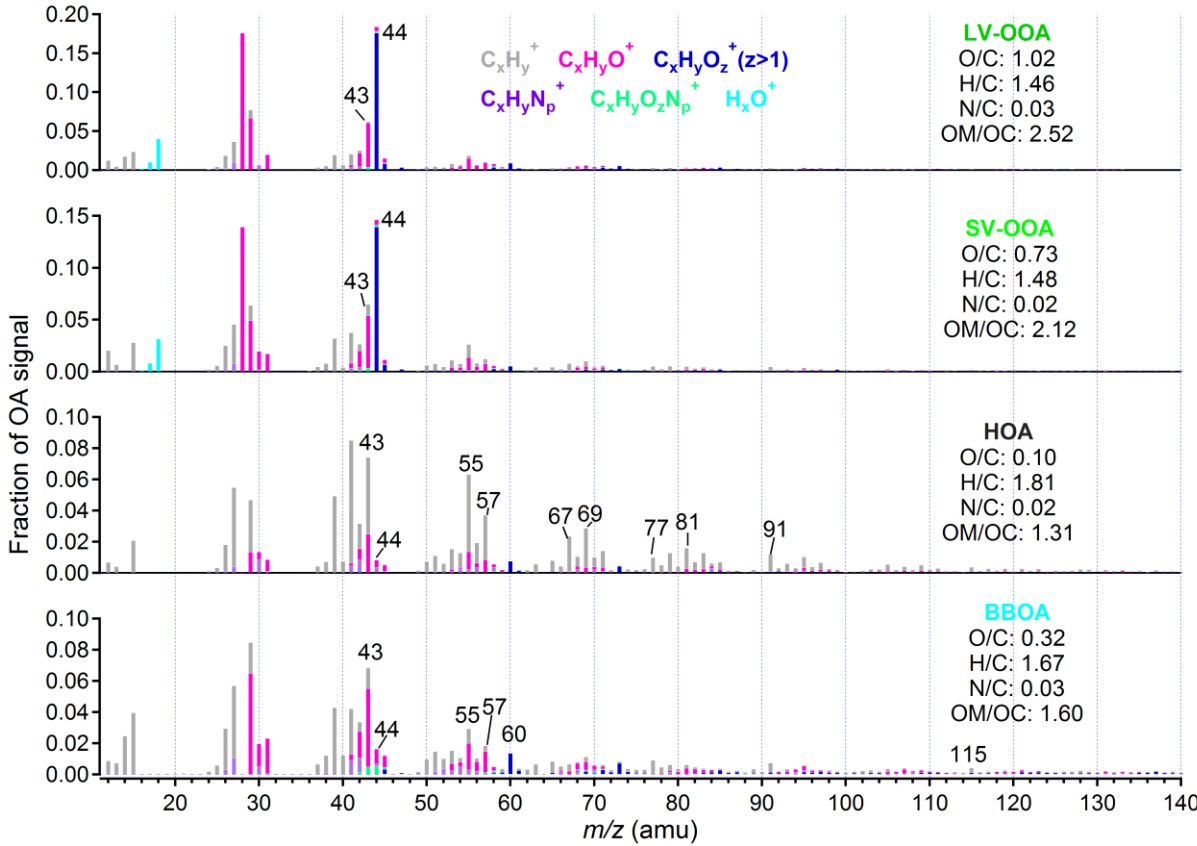

Figure 5. Unit mass spectra of OA factors: LV-OOA, SV-OOA, HOA and BBOA. The elemental ratios and OM/OC ratios of

each component are also added.





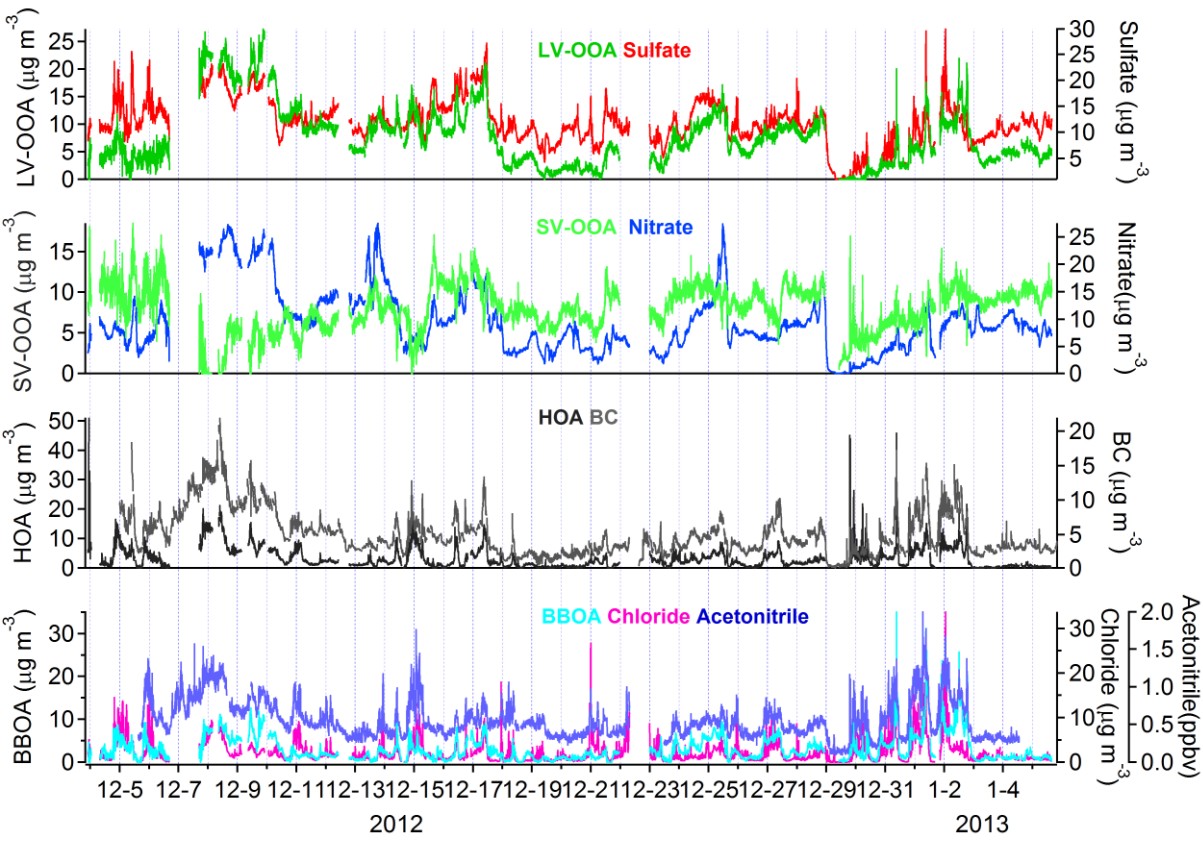

Figure 6. Time series of OA fractions and external tracers (sulfate, nitrate, BC, chloride and acetonitrile).





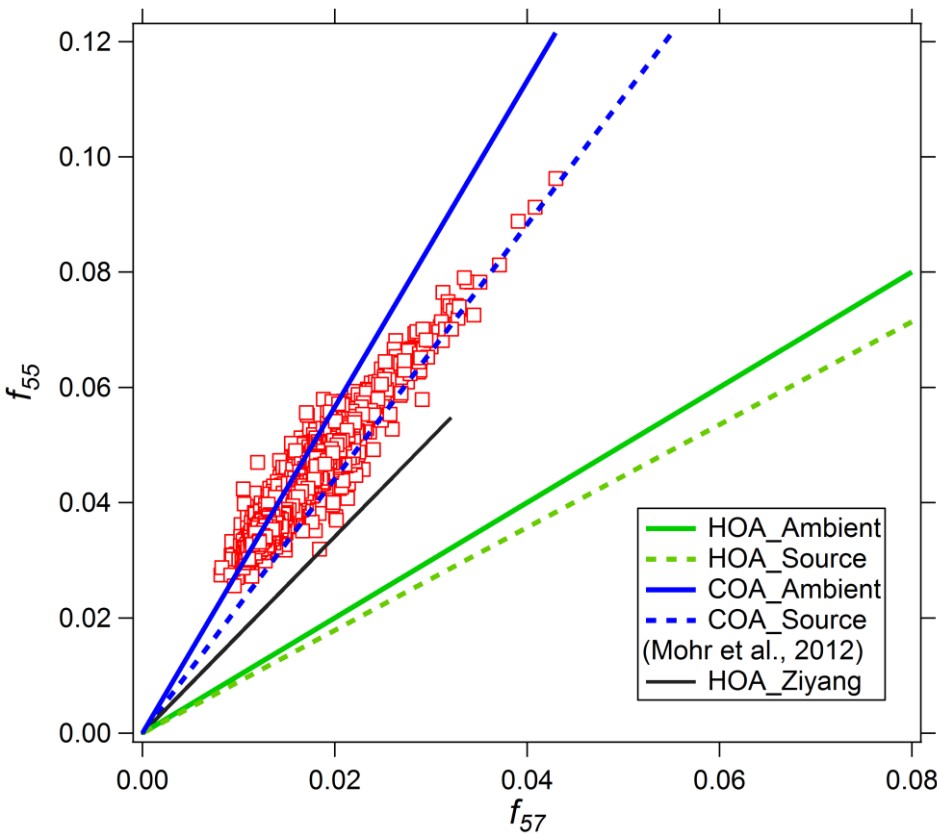

Figure 7. Scatter plot between $f_{55}$ vs. $f_{57}$. The $f_{55}$ vs. $f_{57}$ ratios of "HOA_Ambient" and "COA_Ambient" represent average $f_{55}$ vs. $f_{57}$ values from various PMF HOA and COA factors, and those of "HOA_Source" and "COA_Source" represent $f_{55}$ vs. $f_{57}$ values averaged from several source emission studies reported in Mohr et al. (2012).



Figure 8. (a) Average mass fraction of each OA component. Diurnal variations of concentrations (b) and fractions in OA (c) for different OA components. (d) Fractions of different OA components in total OA (left) depending on OA concentrations and the probability density of OA concentrations (right).



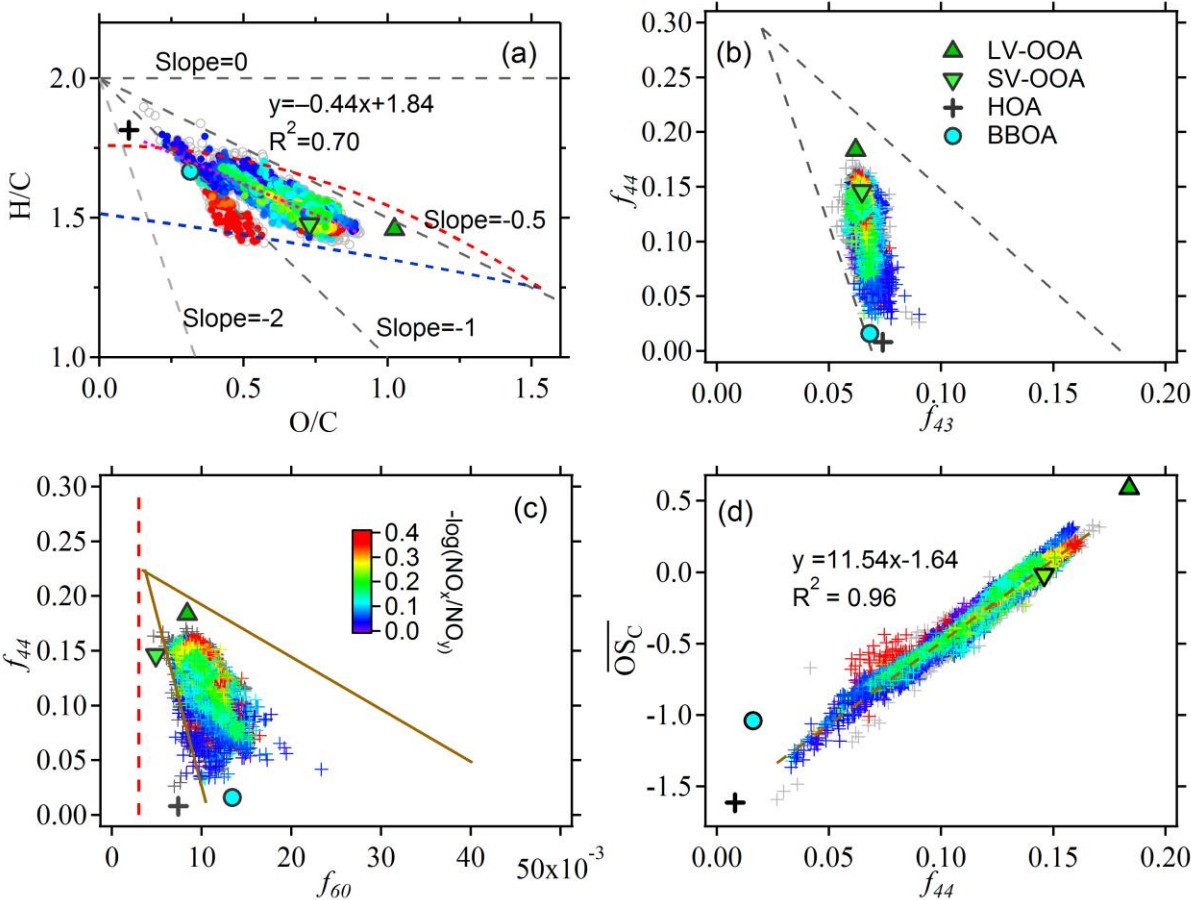

Figure 9. (a) Van Krevelen diagram of organic aerosols. The majority of the data fall into colored triangle lines (Ng et al., 2011). (b) Scattering plot of $f_{44}$ ($m/z$ 44 fraction in organic mass spectra) vs. $f_{43}$. The triangular space defined by dash lines (Ng et al., 2011) indicates the region where the data of OA components fall into. (c) Scattering plot of $f_{44}$ vs. $f_{60}$. The conceptual space for BBOA and the nominal background value at 0.3% (Cubison et al., 2011) are marked by solid and vertical dash lines, respectively. (d) Average carbon oxidation state ($\overline{OS_C}$) vs. $f_{44}$ at Ziyang site. The scattering data points are colored by the photochemical age metric $-\log$ ($NO_x/NO_y$). The locations of OA factors are also marked in all diagrams.





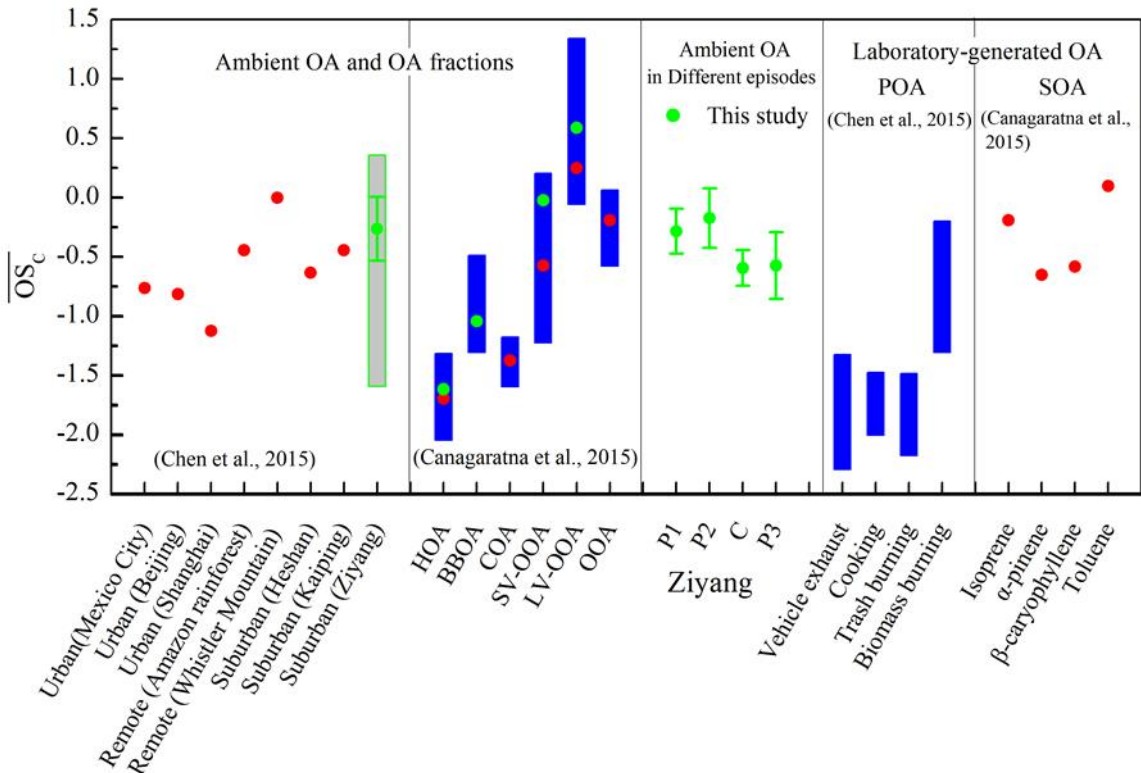

Figure 10. Improved-Ambient results of $\overline{OS_C}$ for OA. The previously reported results are from the summary by Chen et al. (2015) and Canagaratna et al. (2015). The floating bar, dot and error bar mean the range, average and standard deviation of $\overline{OS_C}$.





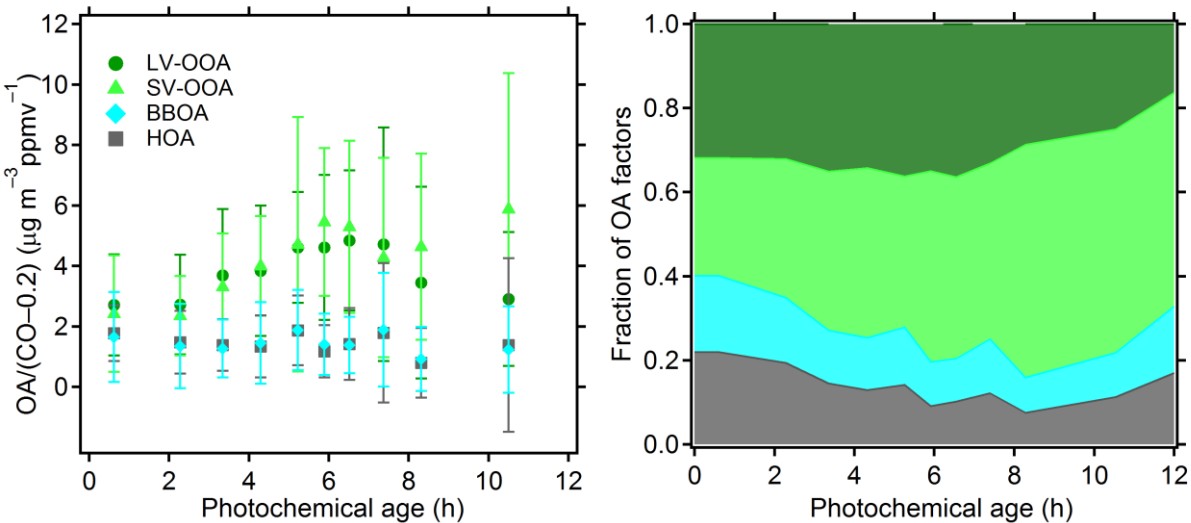

Figure 11. (a) The variations of LV-OOA/ΔCO, SV-OOA/ΔCO, BBOA/ΔCO and HOA/ΔCO with photochemical age. The

dot point and bar in each photochemical age bin are the average value and standard deviation. (b) Mass fraction of each OA

component as a function of photochemical age. The photochemical age is classified into 10 bins by decile of photochemical

5  age.