# Peer review of "Characterization of submicron aerosols influenced by biomass burning at a site in the Sichuan Basin, southwestern China"

_Atmospheric Chemistry and Physics, 2016_

## Referee Comment (RC1)

**General comments:**

The authors present detailed chemical characteristics of atmospheric submicron particles by performing a wintertime field observation at a suburban site in the Sichuan Basin, southwestern China, using multiple advanced instruments such as HR-ToF-AMS, MAAP, GC-MS/FID, PTR-MS, and TEM-EDX. On the basis of AMS high-resolution mass spectra data, four OA factors were identified by PMF source apportionment analysis. Secondary formation and aging process of organic aerosols were also investigated with different approaches; especially, significant influence contributed by biomass burning was discussed.

I would recommend this paper to be accepted after more in depth discussions are included, and after the following specific comments are addressed.

**Specific comments:**

1. The **Introduction** section is not logically connected well with the Results and Discussion part, which mainly focuses on particle chemical characterization, secondary OA formation, aging processes of OA, and the possible influence of biomass burning on these properties. Relating to the abovementioned topics, previous studies and corresponding results, especially in this studied region, were expected to be summarized in the introduction instead of simply included in results and discussion (e.g. Sect. 3.3.4). The introduction described much previous work on poor visibility/haze issues; however the connections with the discussion part were not clearly illustrated. The authors did not explain well why high time resolution particle chemical characterization in Sichuan Basin is needed, and why it is necessary to investigate secondary formation in the influence of biomass burning by using high time resolution aerosol mass

spectra here (**paragraph 2, Page 3**).

**Throughout the paper:** The authors often use the term **biomass burning** organic aerosol (BBOA) to indicate OA contributed by domestic cooking (COA) and residential heating (e.g. Page 3, line 20). On Page 3, line 22-23, biomass burning is actually included in biogenic sources; thus biomass burning could also originate from other open fires such as rice straw burning and forest fires. On Page 4, line 13-15, the authors have pointed out that BBOA and COA are regarded as two different primary sources of OA in some studies, although complete identification among BBOA, COA, and CCOA with PMF analysis is not easy in practice. To avoid confusion during using these different expressions, it would be better to keep a consistent way, and define biomass burning and BBOA of this study clearly in the very beginning.

I would suggest the authors to reorganize the Introduction to better connect with the topics discussed in this manuscript.

2. **Introduction**, Page 4, line 3-5

How should this sentence be understood? Why is primary organic aerosol excluded from the discussion here (only SOA is included)? Is there any difference between OM and OA?

3. **Methodology**, Page 5, line 8

What is the elevation of the sampling site itself? How did you choose the altitude of the starting point of the backward trajectory calculation?

Page 5, line 12, *"Therefore, the atmospheric processes are dominated by the isolated meteorology of the basin."*

Did the backward trajectory analysis really support this idea?

4. Although the chemical composition analysis is focusing on submicron

particles, the actual cut-size of MAAP is $PM_{2.5}$, while that of HR-ToF-AMS is $PM_1$. How did the authors consider this mismatch of two different size ranges into the mass fraction calculation, which could vary significantly with different proportions of BC accounted for $PM_1$? Relevant details should be provided in the measurement and data processing descriptions.

5. Page 7, line 1-2

Why RIE = 4.04 was applied to ammonium? Can you provide any reference or supporting information?

The recommended value of CE = 0.5 is used. Are there any comparative results or strong supporting evidence to verify its applicability? Middlebrook et al. (2011) have demonstrated the composition-dependence of CE with field measurements, suggesting that CE could be higher when ambient particles are composed of high fraction of ammonium nitrate or strongly acidic sulfate. This phenomenon could be significant especially under high RH conditions. In this sense, these issues may need to be considered, as high RH conditions were frequently observed during the investigation period of this work.

6. **Results and discussion**, Page 9, line 4

Is this sentence describing inorganic species or organic species?

7. Page 9, line 7

*"However, the humid air caused by the precipitation may favor the aqueous-phase secondary formation and hygroscopic growth of SNA in turn."*

How should the reader understand "and hygroscopic growth of SNA in turn" here? Please provide some necessary supporting information and illustrate the connections clearly.

8. **Sect. 3.1.2**, Page 10, line 1

*"It was consistent with the morphology and mixing state of single particles, mostly spherical and in internal mixing state (Fig. 4a-d)."*

It is not clear to me if Fig.4 really supports the authors' conclusion. At least, some aggregated soot particles can be clearly seen. Besides, the size resolution of TEM images is 2 µm, much larger than submicron or even ultrafine size ranges.

9. Page 10, line 4

*"… indicating that the aerosols at Ziyang site may be **more aged** than in other areas."* How did you arrive at this conclusion? The higher peak sizes only demonstrate that particles are larger.

10. **Sect. 3.2.1**, Page 11, line 10

*"The MS of HOA correlated well with the average MS of HOA factor reported in previous studies, as well as that of COA, BBOA and vehicle emitted OA (Vehicle-OA) factors (Table S3). Thus, it was likely that the HOA factor was a mixture of COA and other primary organic aerosols."*

In this case, could it be possible to resolve different factors better by increasing the number of factors for the PMF analysis? It is hard to believe that emissions from the three sources (COA, BBOA, and Vehicle-OA) correlate well all the time. The HR-ToF- AMS simply observes fragments. Is there any possibility that there was a specific source of OA during the observation in that area? m/z 60 exists in the HOA factor. Where does it come from: coal combustion or biomass burning?

11. **Sect. 3.2.2**, Page 12, line 13

*"… presented good correlations with BC and acetaldehyde (Table S4), which were mainly emitted from **primary sources**."*

Can you tell that it is only emitted from biomass burning, or is it also contributed by other types of primary sources?

12. **Sect. 3.2.3**, Page 12, line 18

*"… as the influence of biomass burning is **negligible**."*

Applicability of this assumption depends on characteristics of specific observation site, even though some studies have suggested insignificant influences of biomass burning on OA. The authors have also highlighted that BBOA contributes significantly to their data. Accordingly, this concept may not be justified in this study.

13. Page 13, line 4

*"In this study, LV-OOA **correlated well** with SNA (r=0.66-0.68)"*.

Is the reported ***r* value** considered as an indication of good correlation?

14. Page 13, line 8

*"LV-OOA also showed a similar trend to BC (r=0.75), **maybe because** BC was difficult to diffuse and mixed well in the static air."*

The statement is confusing and ambiguous. How should readers understand it?

15. **Sect. 3.3.4**, Page 19, line 2

*"The increase of OA …, **which** was approximate to that reported at Changdao Island …"*

Does the "which" mean the slope of increased OA or contribution of SOA?

16. Page 19, line 5

*"… the average OA/ΔCO ratio decreased with photochemical age, caused by*

*the decrease of LV-OOA/ΔCO ratio."* Is it still valid if the *SV-OOA/ΔCO ratio* increased at the same time?

The following descriptions of the subsequent sentence are unclear. Please clarify them so that the readers can understand it clearly. Namely, how should the readers understand the "**relatively stable** SV-OOA concentrations" resulted from "inhibited evolution from POA to SV-OOA", while "inhibited evolution from POA to LV-OOA resulting in **lower** LV-OOA"?

17. Page 19, line 10

*"SOA dominated OA (56-84%) in both **fresh** and aged plumes…"*

Do you need to define the "fresh" plume in this work to distinguish it from "aged" ones, or provide a certain threshold value in terms of different photochemical ages?

18. Page 19, line 15

*"… implying that the photochemical formation of SV-OOA was more efficiently than that of LV-OOA in this campaign."*

Is this conclusion applicable only to cases for longer photochemical age? We can find from Fig.11 that the fractions of SV-OOA are not always higher than that for LV-OOA, especially when the photochemical age is less than about 6h.

19. Figure 1

In addition to wind speed, wind direction is also an important indicator of air mass origin or possible influence by transportation. Perhaps you can try to display both wind speed and direction parameters in Fig.1(a) and discuss accordingly.

20. The whole passage is generally well organized; however some important

statistics of chemical information are expected to be presented in the manuscript, instead of the supplementary materials. For example, **Table S1** actually contains many new and interesting primary results obtained from this study. The mass concentrations of BC under different meteorological conditions could also be a good case. The contribution of BC to $PM_1$ has been included in abstract and conclusion sections, indicating the importance of BC in chemical characteristics of submicron particles. The corresponding results would be more straightforward to readers if shown in the manuscript.

**Technical corrections:**

1. **Introduction**, Page 2, line 9

   *"… has become one of the most polluted regions in China."*

   Corresponding references are needed, as well as for the specific values that are not obtained from this study (e.g., Page 3, line 17 and 19).

2. **Methodology**, Page 7, line 13

   *"… the diurnal patterns of different factors, etc. (Zhang et al., 2011)"*.

   Please specify the "etc" clearly.

3. **Sect. 3.2**, Page 10

   *"**SOA** (OOA) dominated in OA as much as 71% …"*

   *"… secondary formation (**SOA**+SNA) …"*

   Please be careful when using SOA and OOA, as OOA is not completely the same as SOA.

4. **Sect. 3.3.4**, Page 19, line 21

   *"... and reached saturation frequently (Table S1)."*

   Does it mean average RH or RH?

*Reference: Middlebrook, A. M., Bahreini, R., Jimenez, J. L., and Canagaratna, M. R.: Evaluation of Composition-Dependent Collection Efficiencies for the Aerodyne Aerosol Mass Spectrometer using Field Data, Aerosol. Sci. Tech., 46, 258–271, doi:10.1080/02786826.2011.620041, 2011.*

---

## Referee Comment (RC2)

Hu et al. reported the HR-ToF-AMS results at an urban downwind site in Sichuan basin in winter. The chemical composition and size distributions of submicron aerosols were characterized, and the sources of OA were investigated by PMF. The authors also studied the aging of OA using various approaches, e.g., Van Krevelen diagram, f44 vs. f43, oxidation states, and OA/CO, etc. This study is helpful to understand aerosol variations and oxidation states in southwestern China. However, the English writing is poor, and I often missed the logic when reading this manuscript. In addition, the data quality needs to be further validated and some data interpretations are not convincing. A major revision is needed.

Comments:

1. I raised the same comments in my first review of this manuscript. My major concern is the PMF results. Although the author expanded the PMF results, I still didn't see PMF diagnostic plots for other solutions except Table S2 with a simple description of the reasons. In addition, I don't understand why the authors didn't use the VOCs measurements to evaluate the PMF results.

2. The authors assumed that aerosol particles were neutralized and then got a RIE = 4.04 for ammonium. How are the authors sure that aerosol particles were neutral? Why didn't the authors use pure ammonium nitrate particles from IE calibration data to get the RIE of ammonium?

3. The authors claimed several times the unique of this study "unique geographical and meteorological conditions", particularly in the abstract. But I didn't see the details for this uniqueness (the authors didn't describe it either except high relative humidity).

4. The authors emphasized "influenced by biomass burning" in the title, however this manuscript appears to miss this point in both abstract and text.

---

## Author Comment (AC1) · 16 Aug 2016

**Point-to-Point Responses to Referees' Comments on "Characterization of submicron aerosols influenced by biomass burning at a site in the Sichuan Basin, southwestern China"**

Wei Hu, Min Hu[*], Wei-Wei Hu[#], Hongya Niu, Jing Zheng, Yusheng Wu, Wentai Chen, Chen Chen,

Lingyu Li, Min Shao, Shaodong Xie, Yuanhang Zhang

State Key Joint Laboratory of Environmental Simulation and Pollution Control, College of Environmental Sciences and Engineering, Peking University, Beijing 100871, China

[#]now at: Cooperative Institute for Research in Environmental Sciences, University of Colorado, Boulder, CO 80309, USA

[*]*Correspondence to:* M. Hu (minhu@pku.edu.cn)

**Referee #1**

Hu et al. reported the HR-ToF-AMS results at an urban downwind site in Sichuan basin in winter. The chemical composition and size distributions of submicron aerosols were characterized, and the sources of OA were investigated by PMF. The authors also studied the aging of OA using various approaches, e.g., Van Krevelen diagram, $f_{44}$ vs. $f_{43}$, oxidation states, and OA/CO, etc. This study is helpful to understand aerosol variations and oxidation states in southwestern China. However, the English writing is poor, and I often missed the logic when reading this manuscript. In addition, the data quality needs to be further validated and some data interpretations are not convincing. A major revision is needed.

**Response**: Thanks very much for Referee' comments. We carefully checked and corrected the English again with the help of a native English speaker, and provided more supplementary materials for further validation of data, and give point-to-point responses to Referee's comments and corrected the manuscript accordingly. Please refer to the responses below and revised manuscript.

Comments:

1. I raised the same comments in my first review of this manuscript. My major concern is the PMF results. Although the author expanded the PMF results, I still didn't see PMF diagnostic plots for other solutions except Table S2 with a simple description of the reasons. In addition, I don't understand why the authors didn't use the VOCs measurements to evaluate the PMF results.

**Response**: In the last revised manuscript for ACPD, we added PMF **diagnostic plots** and related tables in **Sect. S3** in the supplementary material.

In this revised manuscript, we added the result of 5-factor solution in the supplementary material. As mentioned in **Table S1** (Table S2 in the last version), SV-OOA and HOA in 4-factor solution were split into three factors with similar spectra, however, different time series (**Fig. S4-S6**). These factors can't be identified definitely. So we select 4-factor as the optimum solution.

[Figure]

**Figure S4.** Unit mass spectra of OA factors for 5-factor solution. SV-OOA and HOA for four-factor solution were split into three factors with similar spectra (Fig. S6), marked as SV-OOA, HOA, and HOA-SV-OOA. The other two are marked as LV-OOA and BBOA. The elemental ratios and OA/OC ratios of each component are also added.

[Figure]

**Figure S5.** Time series of OA fractions for five-factor solution (marked as SV-OOA, HOA, HOA-SV-OOA, LV-OOA and BBOA) and external tracers (sulfate, nitrate, BC, and acetonitrile).

[Figure]

**Figure S6.** Correlation of time series and unit mass spectra of OA factors for 5-factor solution.

As the uncentered correlations shown in Table S2, the MS of OA factors resolved in this study and the average MS of reported OA factors correlated well. HOA resolved in this study, as a mixed factor of vehicle, cooking and other primary emissions, the MS of which correlated well with the average MS of HOA, BBOA, COA, and Vehicle-OA factors reported in previous studies (Table S2).

In the last revised manuscript, we listed the correlation coefficients of OA factors with acetaldehyde and acetonitrile in Table S3. In this revised manuscript, we added the correlation coefficients of OA factors with another three VOC species, toluene, benzene, and acetone. The primary emitted VOC species show good correlation with POA components (HOA and BBOA). In Page 12, Line 3, "*toluene, and benzene*" was added. More details on the VOC characterization and biomass burning contribution to ambient VOCs can be found in Li et al. (2014).

According to all these evaluation, the PMF results are considered to be reasonable.

**Table S3** Correlation coefficients (Pearson's R) of OA factors with gaseous and aerosol species. Correlation coefficients higher than 0.60 are in bold.

| | LV-OOA | SV-OOA | HOA | BBOA |
|---|---|---|---|---|
| $SO_4^{2-}$ | **0.65** | 0.36 | 0.26 | 0.30 |
| $NO_3^-$ | **0.66** | 0.31 | 0.15 | 0.21 |
| $NH_4^+$ | **0.68** | 0.28 | 0.36 | 0.34 |
| $Cl^-$ | 0.22 | -0.08 | 0.57 | 0.49 |
| BC | **0.75** | 0.18 | **0.73** | **0.77** |
| $C_2H_4O_2^+$ | **0.77** | 0.28 | **0.80** | **0.85** |
| $SO_2$ | 0.10 | 0.09 | 0.39 | 0.44 |
| $NO_x$ | 0.31 | -0.13 | **0.62** | 0.47 |
| $NO_y$ | 0.39 | -0.09 | **0.64** | 0.51 |
| $O_3$ | -0.31 | 0.08 | -0.32 | -0.21 |
| CO | 0.20 | -0.09 | 0.49 | 0.42 |
| Acetaldehyde | 0.34 | 0.26 | **0.65** | **0.77** |
| Acetonitrile | 0.44 | -0.02 | **0.73** | **0.68** |
| Toluene | 0.57 | -0.39 | **0.78** | 0.52 |
| Benzene | 0.55 | -0.28 | **0.76** | 0.58 |
| Acetone | 0.48 | 0.14 | 0.49 | 0.54 |
| LV-OOA | **1.00** | | | |
| SV-OOA | 0.37 | **1.00** | | |
| HOA | 0.53 | 0.01 | **1.00** | |
| BBOA | 0.55 | 0.25 | **0.78** | **1.00** |

2. The authors assumed that aerosol particles were neutralized and then got a RIE = 4.04 for ammonium. How are the authors sure that aerosol particles were neutral? Why didn't the authors use pure ammonium nitrate particles from IE calibration data to get the RIE of ammonium?

**Response:** As the last response said, in this study we missed the measurenment of ammonium under MS mode. But the RIE of ammonium under BFSP mode was inaccurate, hence there was no data of ammonium nitrate was available for RIE determination based on AMS. According to the ion balance of filter based results, $PM_{2.5}$ was nearly neutral (**Fig. R1**). The predicted $NH_4^+$ was calculated assuming full neutralization of particulate anions of $NO_3^-$, $SO_4^{2-}$ and $Cl^-$. The measured $NH_4^+$ was less than the predicted $NH_4^+$ (**Fig. R2**), which was mainly caused by the neutralization of $Ca^{2+}$, $Mg^{2+}$ and $Na^+$ in larger sizes (Guo et al., 2010).

Biomass burning can contribute abundant $K^+$ to ambient fine particles. The average concentration of $K^+$ in $PM_{2.5}$ was 1.22 μg m$^{-3}$ during the same campaign. Assuming all $K^+$ existed in submicron size range and $PM_1$ was neutral, the uncertainty of measured $NH_4^+$ mass concentration caused by $K^+$ was 6.7% (**Fig. R3**). Assuming $PM_1$ was neutral, and $K^+$ and other crustal ions were ignorable in submicron size range, as RIE=4.04, the predicted and measured $NH_4^+$ exhibited good consistency with a slope of 0.999 (**Fig. R4**). When the RIE=4.04 was used for ammonium quantification in this study, the uncertainty of measured $NH_4^+$ mass concentration caused by $K^+$ was 6.7%.

In the revised manuscript, "*except for ammonium for which RIE=4.04 was used assuming $PM_1$ was neutral and other metallic species were ignorable in $PM_1$ (water-soluble $K^+$, $Ca^{2+}$, $Mg^{2+}$, and $Na^+$ were 1.22, 0.25, 0.04, and 0.21 μg m$^{-3}$ in $PM_{2.5}$).*" was added in **Page 7, Line 16**.

[Figure]

Figure R1. Scatter plots between total cation and anion concentrations in PM2.5 filter samples.

[Figure]

Figure R2. Scatter plots between measured NH4+ and predicted NH4+ of PM2.5 filter samples.

[Figure]

Figure R3. Scatter plots between AMS measured $NH_4^+$ and $K^+$, and predicted $NH_4^+$ mole concentration.

[Figure]

Figure R4. Scatter plots between AMS measured $NH_4^+$ and predicted $NH_4^+$.

3. The authors claimed several times the unique of this study "unique geographical and meteorological conditions", particularly in the abstract. But I didn't see the details for this uniqueness (the authors didn't describe it either except high relative humidity).

**Response:** The "unique geographical and meteorological conditions" in the Sichuan Basin was addressed as "*High emissions of gaseous and particulate pollutants, such as volatile organic compounds (VOCs), SO₂, organic carbon (OC), black carbon (BC) and fine particles (PM₂.₅), are found in the Sichuan Basin over China (He, 2012). Adversely influenced by the particular topographic condition, the Sichuan Basin is within the region of the lowest wind speed and relatively high humidity over China all year round (Chen and Xie, 2013; Yang et al., 2011). The highest annual mean aerosol optical depth (AOD) in the Sichuan Basin from 2000 to 2010 across China reflected the importance of large topography in aerosol accumulation (Luo et al., 2013).*" in Paragraph 1 in the Introduction.

The topography of the Sichuan Basin is shown in Fig. S1 in the supplementary material. The backward trajectory clusters of air masses during the observation periods illustrated in **Fig. 1** showed that "*the atmospheric processes were dominated by the isolated meteorology of the basin*" (**Page 5, Line 21**).

All these gave the details for the "*unique geographical and meteorological conditions*".

4. The authors emphasized "influenced by biomass burning" in the title, however this manuscript appears to miss this point in both abstract and text.

**Response:** In the abstract, we mentioned that "*During the episode obviously influenced by primary emissions, the contributions of BBOA to OA (26%) and PM₁ (11%) were much higher than those (10-17%, 4-7%) in the clean and other polluted episodes, highlighting the significant influence of biomass burning.*"

In the Paragraph 2 in the Introduction, we summarized the contribution of biomass burning to aerosol pollution in the Sichuan Basin, and explained why "*it is necessary to investigate secondary formation in the influence of biomass burning by using high time resolution aerosol mass spectra*".

For the Results and discussion Sect., we resolved the BBOA component of OA, and investigated the secondary formation in the influence of biomass burning (Paragraph 3 in Sect. 3.2.3, Paragraph 3 in Sect.3.3.2, and Sect. 3.3.4).

Thank you very much for your helpful comments. Your any further comments and suggestions are appreciated.

**Referee #2**

**General comments:**

The authors present detailed chemical characteristics of atmospheric submicron particles by performing a wintertime field observation at a suburban site in the Sichuan Basin, southwestern China, using multiple advanced instruments such as HR-ToF-AMS, MAAP, GC-MS/FID, PTR-MS, and TEM-EDX. On the basis of AMS high-resolution mass spectra data, four OA factors were identified by PMF source apportionment analysis. Secondary formation and aging process of organic aerosols were also investigated with different approaches; especially, significant influence contributed by biomass burning was discussed.

I would recommend this paper to be accepted after more in depth discussions are included, and after the following specific comments are addressed.

**Response:** Thanks for Reviewer's comment. We have revised the paper according to Referee's comments to improve the quality of this manuscript. Please see the detailed response below and changes marked in blue in the revised manuscript.

**Specific comments:**

1. The **Introduction** section is not logically connected well with the Results and Discussion part, which mainly focuses on particle chemical characterization, secondary OA formation, aging processes of OA, and the possible influence of biomass burning on these properties. Relating to the abovementioned topics, previous studies and corresponding results, especially in this studied region, were expected to be summarized in the introduction instead of simply included in results and discussion (e.g. Sect. 3.3.4). The introduction described much previous work on poor visibility/haze issues; however the connections with the discussion part were not clearly illustrated. The authors did not explain well why high time resolution particle chemical characterization in Sichuan Basin is needed, and why it is necessary to investigate secondary formation in the influence of biomass burning by using high time resolution aerosol mass spectra here (**paragraph 2, Page 3**).

**Throughout the paper:** The authors often use the term **biomass burning** organic aerosol (BBOA) to indicate OA contributed by domestic cooking (COA) and residential heating (e.g. Page 3, line 20). On Page 3, line 22-23, biomass burning is actually included in biogenic sources; thus biomass burning could also originate from other open fires such as rice straw burning and forest fires. On Page 4, line 13-15, the authors have pointed out that BBOA and COA are regarded as two different primary sources of OA in some studies, although complete identification among BBOA, COA, and CCOA with PMF analysis is not easy in practice. To avoid confusion during using these different expressions, it would be better to keep a consistent way, and define biomass burning and BBOA of this study clearly in the very beginning.

I would suggest the authors to reorganize the Introduction to better connect with the topics discussed in this manuscript.

**Response:** We reorganized the Introduction according to Referee's suggestions.

In the revision, the content in Sect. 3 "*The aging process plays an important role in the life cycle of atmospheric aerosols. The chemical composition, hygroscopicity and solubility of atmospheric aerosols are varied due to the aging process, which not only influences their optical properties and capability to form clouds, but also changes their impacts on environment, climate and human health. The aging process of OA can be characterized by some metrics and tools, including C/H/O atomic ratios, van Krevelen diagram (VK diagram), OA/OC ratio, OA/ΔCO, average carbon oxidation state ($\overline{OS_C}$), and the abundance of characteristic fragment ions ($f_{43}$ and $f_{44}$), etc., to provide the basis for model simulation of SOA formation (de Gouw and Jimenez, 2009; Kroll et al., 2011; Hu et al., 2013).*" was moved into the Introduction.

The contents of previous work on poor visibility/haze issues in the Introduction "*The Sichuan Basin has suffered from long-term poor visibility since the 1970s (Chen and Xie, 2012, 2013). The visibility degradation primarily results from anthropogenic pollutants and synoptic processes. Anthropogenic aerosols and moisture at the surface are the dominant determinants of the AOD, and the spatial distributions of both AOD and light extinction coefficient ($B_{ext}$) are strongly influenced by regional*

topography (Wang et al., 2013)." "Severe visibility deterioration and frequent hazy days have become vital concerns in the Sichuan Basin." were deleted.

In the text, we mentioned that "*variability of fine particle concentrations and physiochemical characteristics serves as another crucial factor in explaining the degradation of air quality in the Sichuan Basin*", "*Though several published papers focused on aerosol chemical and physical properties in the Sichuan Basin, highly time-resolved studies are rarely conducted*". In addition, according to previous studies, biomass burning contributes importantly to air pollution in the Sichuan Basin (Wang et al., 2013; Yang et al., 2011), and secondary pollutants from biomass burning significantly influence local and regional air quality, chemical processes, and even climate change (Niu et al., 2016). Therefore, "*high time resolution particle chemical characterization in Sichuan Basin is needed*", and "*it is also necessary to investigate secondary formation in the influence of biomass burning by using high time resolution aerosol mass spectra*".

In the revision, we added "***Further, biomass burning contributes importantly to air pollution in the Sichuan Basin (Wang et al., 2013; Yang et al., 2011). Secondary formation from biomass burning emissions can significantly influence local and regional air quality, atmospheric processes, and even climate change (Niu et al., 2016).***" in Page 3, Line 11.

We agree with that "*biomass burning could also originate from other open fires such as rice straw burning and forest fires*", but according to the season (winter) of this study and vegetation cover in the Sichuan Basin, house heating with wood and straw is much more common (*Wang et al., 2013; Yang et al., 2011*) than "*rice straw burning and forest fires*". Based on previous studies, COA resolved by AMS-PMF analysis, refers to OA emitted by food during cooking activities, with no relation to fuels for cooking (Allan et al., 2010).

In the revision, "***Noted that COA refers to OA emitted by food during cooking activities, with no relation to fuels for cooking (Allan et al., 2010).***" was added in Page 4, Line 15.

To avoid confusion, Page 3, Line 20, "*biomass burning emissions via residential cooking and heating*" was changed into "***biomass burning as residential fuels***".

Page 12, Line 8, "*cooking emissions*" was changed into "***COA***".

Page 12, Line 10, *"Biomass burning via cooking and house heating"* was changed into *"**Residential biomass burning**"*.

2. **Introduction**, Page 4, line 3-5

How should this sentence be understood? Why is primary organic aerosol excluded from the discussion here (only SOA is included)? Is there any difference between OM and OA?

**Response:** In the revision, *"**Many studies refer to the particulate organic matter (OM)**"* was revised into *"**Many studies refer to secondary organic aerosols (SOA)**"*. Particulate organic matter and organic aerosol have the same meaning. To avoid confusing, in the whole manuscript, "OM" (abbreviated form of organic matter or organic mass) was replaced by "OA".

3. **Methodology**, Page 5, line 8

What is the elevation of the sampling site itself? How did you choose the altitude of the starting point of the backward trajectory calculation?

Page 5, line 12, *"Therefore, the atmospheric processes are dominated by the isolated meteorology of the basin."*

Did the backward trajectory analysis really support this idea?

**Response:** The elevation range of the Sichuan Basin bottom is about 200-800 m (Ziyang, 300-550 m), and the basin was surrounded by plateaus and mountains at the elevation of 2000-3000 m. However, the altitude of the starting point of the backward trajectory calculation **has no relation to the elevation of the sampling site**, and it is the height above ground level (AGL). The altitude of 500 m-AGL was selected as an approximation of the well-mixed boundary layer (Huang et al., 2010; Lu et al., 2012). So the backward trajectory analysis could support that *"the atmospheric processes are dominated by the isolated meteorology of the basin."*

In the revision, *"**above ground level**"* was added in Page 5, L17.

4. Although the chemical composition analysis is focusing on submicron particles, the actual cut-size of MAAP is $PM_{2.5}$, while that of HR-ToF-AMS is $PM_1$. How did the authors consider this mismatch

of two different size ranges into the mass fraction calculation, which could vary significantly with different proportions of BC accounted for $PM_1$? Relevant details should be provided in the measurement and data processing descriptions.

**Response:** Ambient BC particles are largely found in the Aitken and accumulation modes (i.e., in the submicron range) because of their formation mechanism (Bond et al., 2013; Huang et al., 2012a; Rose et al., 2006). The sum of non-refractory species measured by HR-ToF-AMS and BC measured by instruments such as MAAP or aethalometer with the cut-size of 2.5 μm is often treated as total $PM_1$ in previous studies (Huang et al., 2010, 2012b, 2013; He et al., 2011; Hu et al., 2013, 2016). In this study, the morphology of individual particles also indicated that the sizes of soot particles were less than 1 μm (Fig. 4). So we thought this match has little influence on $PM_1$.

In the revision, "***Atmospheric BC particles are mostly in the Aitken and accumulation modes (i.e., in the submicron range) because of their formation mechanisms (Bond et al., 2013).***" was added in Page 6, Line 16.

5. Page 7, line 1-2

Why RIE = 4.04 was applied to ammonium? Can you provide any reference or supporting information?

The recommended value of CE = 0.5 is used. Are there any comparative results or strong supporting evidence to verify its applicability? Middlebrook et al. (2011) have demonstrated the composition-dependence of CE with field measurements, suggesting that CE could be higher when ambient particles are composed of high fraction of ammonium nitrate or strongly acidic sulfate. This phenomenon could be significant especially under high RH conditions. In this sense, these issues may need to be considered, as high RH conditions were frequently observed during the investigation period of this work.

**Response:** The RIE=4.04 was used for ammonium quantification in this study by assuming $PM_1$ was neutral, and $K^+$ and other crustal ions were ignorable in submicron size range. That is, as RIE=4.04, the predicted and measured $NH_4^+$ exhibited good consistency with a slope of 0.999, and the uncertainty of measured $NH_4^+$ mass concentration caused by $K^+$ (assuming all $K^+$ existed in submicron size range

and PM$_1$) was neutral was 6.7%. Please refer to the response to Comment 2 of Referee #1 for more details.

In the revised manuscript, "*except for ammonium for which RIE=4.04 was used assuming PM$_1$ was neutral and other metallic species were ignorable in PM$_1$ (water-soluble K$^+$, Ca$^{2+}$, Mg$^{2+}$, and Na$^+$ were 1.22, 0.25, 0.04, and 0.21 $\mu g\ m^{-3}$ in PM$_{2.5}$).*" was added in Page 7, Line 16.

The chemical composition-based estimation of CE was estimated following the method addressed in Middlebrook et al. (2012). Only in the morning and afternoon on 13 December, the calculated CE reached to 0.51 for several hours due to high fraction of ammonium nitrate. The calculated CE showed little variations dependent on aerosol acidity and fraction of ammonium nitrate (mostly 0.45). Therefore, in order to compare with previous studies, here the recommended value of CE = 0.5 was used.

In the revision, "*All algorithm results of AMS collection efficiencies (CE) based on aerosol chemical compositions and sampling line RH (Middlebrook et al., 2012) were approximately 0.5*." was added in Page 7, Line 11.

6. **Results and discussion**, Page 9, line 4

Is this sentence describing inorganic species or organic species?

**Response:** This sentence "*Since the secondary compositions were dominant in PM$_1$…*" is describing both inorganic and organic species. As described above, the main chemical compositions, SNA accounted for about 49%, and OA accounted for about 36% of PM$_1$, to which SOA should contribute a part (we haven't mentioned it until Sect. 3.2). So here we described that "*the secondary compositions were dominant in PM$_1$*".

7. Page 9, line 7

"*However, the humid air caused by the precipitation may favor the aqueous-phase secondary formation and hygroscopic growth of SNA in turn.*"

How should the reader understand "and hygroscopic growth of SNA in turn" here? Please provide some necessary supporting information and illustrate the connections clearly.

**Response:** In the revision, "*However, the humid air caused by the precipitation may favor the aqueous-phase secondary formation and hygroscopic growth of SNA in turn*" was changed into "***However, the humid air after the precipitation may favor the aqueous-phase secondary formation and hygroscopic growth of SNA, causing the stable or even slightly increased concentration of SNA (Fig.2)***".

8. **Sect. 3.1.2**, Page 10, line 1

*"It was consistent with the morphology and mixing state of single particles, mostly spherical and in internal mixing state (Fig. 4a-d)."*

It is not clear to me if Fig.4 really supports the authors' conclusion. At least, some aggregated soot particles can be clearly seen. Besides, the size resolution of TEM images is 2 μm, much larger than submicron or even ultrafine size ranges.

**Response:** According to **Fig. 4**, the equivalent diameters (Niu et al., 2011, 2012) of most particles were in submicron range. Most particles were spherical and in internal mixing state. Aggregated soot particles only accounted for a very small part because fresh soot particles in chain or aggregate shape could be modified into core-shell structure rapidly in the atmosphere (Niu et al., 2011, 2012). The individual particle analysis is being further conducted and will be prepared to be published.

9. Page 10, line 4

*"... indicating that the aerosols at Ziyang site may be **more aged** than in other areas."* How did you arrive at this conclusion? The higher peak sizes only demonstrate that particles are larger.

**Response:** *"... indicating that the aerosols at Ziyang site may be more aged than in other areas."* was revised into *"**…implying that the aqueous reactions under the high RH condition in Ziyang could cause a faster particle growth rate of secondary species than in other areas (Hu et al., 2016).**"*

10. **Sect. 3.2.1**, Page 11, line 10

*"The MS of HOA correlated well with the average MS of HOA factor reported in previous studies, as well as that of COA, BBOA and vehicle emitted OA (Vehicle-OA) factors (Table S3). Thus, it was likely that the HOA factor was a mixture of COA and other primary organic aerosols."*

In this case, could it be possible to resolve different factors better by increasing the number of factors for the PMF analysis? It is hard to believe that emissions from the three sources (COA, BBOA, and Vehicle-OA) correlate well all the time. The HR-ToF-AMS simply observes fragments. Is there any possibility that there was a specific source of OA during the observation in that area? m/z 60 exists in the HOA factor. Where does it come from: coal combustion or biomass burning?

**Response:** We tried to resolve more factors for PMF analysis. While according to the dialogistic parameters, correlation between OA factors and external tracers, and uncentered correlations between the MS of OA factors resolved in this study and the average MS of reported OA factors, the result of four factors was the optimal solution as described in Sect. S1.

As described in the text, "*Factor analysis (e.g., PMF analysis) suffers a limitation, as it is incapable of separating independent sources completely, and the resolved factor may be a mixture of various sources.*" HOA is a surrogate for urban, combustion-related POA except the identified factors, such as BBOA, COA, and CCOA (Zhang et al., 2011).

The resolved HOA factor in this study was considered to be a mixture of other primary emitted OA, such as COA and Vehicle-OA according to the results of correlation analysis (Table S2 and Table S3), but we couldn't conclude that the emissions from primary sources, such as COA, BBOA, and Vehicle-OA, correlated well all the time.

Coal combustion emissions can contribute to *m/z 60* ($C_2H_4O_2^+$) (Aiken et al., 2010; Hu et al., 2016). The low abundances of *m/z 60* ($C_2H_4O_2^+$) in both HOA and BBOA factors were also appeared in previous studies (DeCarlo et al., 2010; He et al., 2011; Hu et al., 2016; Huang et al., 2013; Ulbrich et al., 2012). The low abundances of *m/z 60* were also found in the resolved COA factors by Hu et al. (2016) and Huang et al. (2010). So the *m/z 60* in HOA factor in this study could be from coal combustion or other combustion emission sources.

11. **Sect. 3.2.2**, Page 12, line 13

*"... presented good correlations with BC and acetaldehyde (Table S4), which were mainly emitted from **primary sources**."*

Can you tell that it is only emitted from biomass burning, or is it also contributed by other types of primary sources?

**Response:** We cannot assure that BC and acetaldehyde were only emitted from biomass burning. BC and acetaldehyde can also be contributed by other types of primary sources. So we described that they *"were mainly emitted from primary sources"*.

In the revision, *"which were mainly emitted from primary sources"* was revised into "**which could be partly emitted from biomass burning**".

12. **Sect. 3.2.3**, Page 12, line 18

*"... as the influence of biomass burning is **negligible**."*

Applicability of this assumption depends on characteristics of specific observation site, even though some studies have suggested insignificant influences of biomass burning on OA. The authors have also highlighted that BBOA contributes significantly to their data. Accordingly, this concept may not be justified in this study.

**Response:** Jimenez et al. (2009) reviewed that there is strong evidence that most atmospheric OOA is secondary, and OOA levels are consistent with SOA estimates using other methods. Ng et al. (2011) also concluded that OOA are good surrogates for SOA under most conditions. Herndon et al. (2008) and Lanz et al. (2007) resolved both BBOA and OOA factors in their studies, and also found that increases in OOA are strongly correlated with photochemical activity and other secondary species.

In the revision, "*... as the influence of biomass burning is negligible*" was changed into "**...under most conditions (Jimenez et al., 2009; Ng et al., 2011)**".

13. Page 13, line 4

*"In this study, LV-OOA **correlated well** with SNA (r=0.66-0.68)"*.

Is the reported **r value** considered as an indication of good correlation?

**Response**: The significance of Pearson correlation coefficient "**p<0.01**" was added in Page, Line, and also added after other Pearson correlation coefficients in the whole manuscript.

14. Page 13, line 8

*"LV-OOA also showed a similar trend to BC (r=0.75), **maybe because** BC was difficult to diffuse and mixed well in the static air."*

The statement is confusing and ambiguous. How should readers understand it?

**Response**: In the revision, *"LV-OOA also showed a similar trend to BC (r=0.75), **maybe because** BC was difficult to diffuse and mixed well in the static air"* was changed into *"**LV-OOA also showed a similar trend to BC (r=0.75), because the aged OA can mix well with BC due to the static air in the basin.**"*

15. **Sect. 3.3.4**, Page 19, line 2

*"The increase of OA ..., **which** was approximate to that reported at Changdao Island ..."*

Does the "which" mean the slope of increased OA or contribution of SOA?

**Response**: Herein "which" means the slope of increased OA. In the revision, *"The increase of OA (1.2 μg $m^{-3}$ $ppmv^{-1}$ $h^{-1}$) was almost completely attributed to the contribution of SOA, which was approximate to that reported at Changdao Island (1.3 μg $m^{-3}$ $ppmv^{-1}$ $h^{-1}$) and lower than the ratios (2-5 $ppmv^{-1}$ $h^{-1}$) reported in Mexico City and the US (Hu et al., 2013)."* was changed into *"**The increasing slope of OA (1.2 μg $m^{-3}$ $ppmv^{-1}$ $h^{-1}$), which was approximate to that at Changdao Island (1.3 μg $m^{-3}$ $ppmv^{-1}$ $h^{-1}$) and lower than those (~2-5 $ppmv^{-1}$ $h^{-1}$) in Mexico City and the US (Hu et al., 2013), was almost completely attributed to the contribution of SOA.**"*

16. Page 19, line 5

*"... the average OA/ΔCO ratio decreased with photochemical age, caused by the decrease of LV-OOA/ΔCO ratio."* Is it still valid if the *SV-OOA/ΔCO ratio* increased at the same time?

The following descriptions of the subsequent sentence are unclear. Please clarify them so that the readers can understand it clearly. Namely, how should the readers understand the "**relatively stable** SV-OOA concentrations" resulted from "inhibited evolution from POA to SV-OOA", while "inhibited evolution from POA to LV-OOA resulting in **lower** LV-OOA"?

**Response**: *"… the average OA/ΔCO ratio decreased with photochemical age, caused by the decrease of LV-OOA/ΔCO ratio"* is a description of the data in Fig. 11a. It was still valid in this study as the SV-OOA/ΔCO ratio slightly increased at the same time. In the revision, "*(Fig. 11a)*"was added in Page 19, Line 16.

According to atmospheric observations and laboratory experiments, as photochemistry proceeds, the signature of OA is transformed and the OA spectra become more similar first to that of ambient SV-OOA and then increasingly to that of LV-OOA. Atmospheric oxidation of OA converges toward highly aged LV-OOA regardless of the original OA sources, with the original source signature being replaced by that of atmospheric oxidation (Jimenez et al., 2009). In this study, the intermediate product, SV-OOA, was already dominant in OA with the increase of the photochemical age, consistent with previous observations in the urban, suburban and even rural areas (Jimenez et al., 2009). So the "lower LV-OOA concentrations" can result from "inhibited evolution from POA to LV-OOA", while the SV-OOA concentration maintained relatively stable.

In the revision, "*Atmospheric oxidation of OA converges toward greatly aged LV-OOA despite of the original OA sources (Jimenez et al., 2009).*" was added in Page 19, Line 16.

In Page 19, Line 20, "*because SV-OOA was already dominant in OA (Jimenez et al., 2009)*" was added.

17. Page 19, line 10

*"SOA dominated OA (56-84%) in both **fresh** and aged plumes…"*

Do you need to define the "fresh" plume in this work to distinguish it from "aged" ones, or provide a certain threshold value in terms of different photochemical ages?

**Response**: According to Fig. 11b, OOA dominated OA at different photochemical ages, so here we didn't define the "fresh" plumes to distinguish it from "aged" ones. In the revision, *"SOA dominated OA (56-84%) in both fresh and aged plumes…"* was changed into "**OOA dominated OA (56-84%) in both fresh and aged plumes…"**.

18. Page 19, line 15

*"… implying that the photochemical formation of SV-OOA was more efficiently than that of LV-OOA in this campaign."*

Is this conclusion applicable only to cases for longer photochemical age? We can find from Fig.11 that the fractions of SV-OOA are not always higher than that for LV-OOA, especially when the photochemical age is less than about 6h.

**Response**: **Page 20, Line 1**, "*In aged plumes*" was revised into "**as the photochemical age longer than 6 h (Fig. 11)**".

19. Figure 1

In addition to wind speed, wind direction is also an important indicator of air mass origin or possible influence by transportation. Perhaps you can try to display both wind speed and direction parameters in Fig.1 (a) and discuss accordingly.

**Response**: As mentioned in the manuscript, "*the stagnant air prevailed in the one-month campaign due to the basin terrain*" (Page 5, Line 19) and "*during the whole campaign, calm occurred frequently (Fig. 2a)*" (Page 9, Line 3), so we didn't display the wind direction.

20. The whole passage is generally well organized; however some important statistics of chemical information are expected to be presented in the manuscript, instead of the supplementary materials. For example, **Table S1** actually contains many new and interesting primary results obtained from this study. The mass concentrations of BC under different meteorological conditions could also be a good case. The contribution of BC to $PM_1$ has been included in abstract and conclusion sections, indicating the importance of BC in chemical characteristics of submicron particles. The corresponding results would be more straightforward to readers if shown in the manuscript.

**Response**: Thanks for Referee's Comment. Table S1 was moved into the manuscript as Table 1, and the contents related to Table 1 were revised accordingly.

**Technical corrections:**

1. **Introduction**, Page 2, line 9

 *"... has become one of the most polluted regions in China."*

Corresponding references are needed, as well as for the specific values that are not obtained from this study (e.g., Page 3, line 17 and 19).

**Response**: "*(Chen and Xie, 2012)*" was added after "*... has become one of the most polluted regions in China.*"

"*The concentration of OA in molecular level using GC/MS analysis was extremely high (9.7 μg m$^{-3}$ in winter) in Chongqing because of its active industrialization and urbanization. Anthropogenic sources, such as coal combustion, cooking and vehicle emissions, contributed to OA primarily. Levoglucosan occupied around 90% of total identified sugars in winter (700 ng m$^{-3}$) and summer (123 ng m$^{-3}$). The high levels of levoglucosan were most likely caused by biomass burning emissions via residential cooking and heating, especially in winter (Wang et al., 2006).*" was revised into "***Wang et al. (2006) reported that the concentration of OA identified in molecular level was extremely high (9.7 μg m$^{-3}$) in winter in Chongqing because of its active industrialization and urbanization. Anthropogenic sources, such as coal combustion, cooking and vehicle emissions, contributed to OA primarily. High levels of levoglucosan, most likely emitted from biomass burning as residential fuels, occupied around 90% of total identified sugars (700 ng m$^{-3}$) (Wang et al., 2006).***"

2. **Methodology**, Page 7, line 13

 *"... the diurnal patterns of different factors, etc. (Zhang et al., 2011)".*

Please specify the "etc" clearly.

**Response**: *As described in Zhang et al. (2011), "the interpretability of the OA factors should be evaluated on the basis of their mass spectral features and temporal variation patterns." "The interpretations of the OA factors are usually based on the following considerations:*

*1. the temporal correlations of factors with tracer species representative of specific emissions and processes;*

*2. the mass spectral features of each factor, for example peak distribution patterns, signature fragments, and oxidation state;*

*3. the repetitive temporal or diurnal variation patterns that are indicative of specific human activities or meteorological patterns (for example traffic rush hours, dilution because of the increase of the planetary boundary layer, cooking emissions during mealtimes, photochemical production of secondary species, etc.);*

*4. the estimated size distributions of OA factors (or tracer ions) and their evolution patterns;*

*5. information regarding airmass trajectories and locations of upwind source regions; and*

*6. other collocated observations that enable the isolation of special cases (e.g., new particle formation and growth events identified according to scanning mobility particle sizer measurements and well-defined SOA growth events)."*

In the revision, *"...be defined via comparing ... the diurnal patterns of different factors, etc. (Zhang et al., 2011)"* was revised into *"**...be evaluated via comparing … the diurnal patterns of different factors (Zhang et al., 2011).**"*

3. **Sect. 3.2**, Page 10

*"**SOA** (OOA) dominated in OA as much as 71% ..."*

*"... secondary formation (**SOA**+SNA) ..."*

Please be careful when using SOA and OOA, as OOA is not completely the same as SOA.

**Response**: As mentioned in the response to Comment 12, OOA are good surrogates for SOA under most conditions.

In the revision, *"SOA (OOA) dominated in OA as much as 71% ..."* was revised into *"**OOA dominated in OA as much as 71% …**", "secondary formation (SOA+SNA) contributed to PM$_1$ as high as 76%"* was revised into *"**secondary formation (OOA+SNA) contributed to PM$_1$ as high as 76%**".*

4. **Sect. 3.3.4**, Page 19, line 21

 *"... and reached saturation frequently (Table S1)."*

Does it mean average RH or RH?

**Response**: It means the measured RH, and the measurement time-resolution for meteorological parameters was one minute. In the revision, *"The average RH in Ziyang during the campaign was 80±19% (12-100%), and reached saturation frequently (Table S1)"* was changed into *"**The RH in Ziyang during the campaign was 80±19% (12-100%) on average (Table 1), and reached saturation frequently**".*

Thank you very much for your helpful comments. Your any further comments and suggestions are appreciated.

Organic aerosols (OA) are very significant components in fine particulate matter (Zhang et al., 2007). Several results on the compositions and sources of OA in the Sichuan Basin have also been reported. Wang et al. (2006) found that the concentration of OA in molecular level was extremely high (9.7 µg m$^{-3}$) in winter in Chongqing because of its active industrialization and urbanization. Anthropogenic sources, such as coal combustion, cooking and vehicle emissions, contributed to OA primarily. High levels of levoglucosan that were most likely emitted from residential biomass burning, occupied around 90% of total identified sugars (700 ng m$^{-3}$) (Wang et al., 2006). Li et al. (2013a) drew 
[revised manuscript text omitted]

[Figure]

**Figure S4.** Unit mass spectra of OA factors for 5-factor solution. SV-OOA and HOA for four-factor solution were split into three factors with similar spectra (Fig. S6), marked as SV-OOA, HOA, and HOA-SV-OOA. The other two are marked as LV-OOA and BBOA. The elemental ratios and OA/OC ratios of each component are also added.

[Figure]

**Figure S5.** Time series of OA fractions for five-factor solution (marked as SV-OOA, HOA, HOA-SV-OOA, LV-OOA and BBOA) and external tracers (sulfate, nitrate, BC, and acetonitrile).

[Figure]

**Figure S6.** Correlation of time series and unit mass spectra of OA factors for 5-factor solution.

**Table S2** The uncentered correlation coefficients between the MS of OA factors resolved in this study and the average MS of OA factors.

| | Ziyang | | | | Average | | | | | | |
|---|---|---|---|---|---|---|---|---|---|---|---|
| | LV-OOA | SV-OOA | HOA | BBOA | LV-OOA | SV-OOA | HOA | BBOA | CCOA | COA | Vehicle-OA |
| **LV-OOA** | 1.00 | | | | 0.99 | 0.92 | | | | | |
| **SV-OOA** | 0.98 | 1.00 | | | 0.99 | 0.97 | | | | | |
| **HOA** | 0.38 | 0.52 | 1.00 | | | | 0.96 | 0.89 | 0.65 | 0.96 | 0.88 |
| **BBOA** | 0.48 | 0.59 | 0.88 | 1.00 | | | 0.86 | 0.93 | 0.65 | 0.81 | 0.68 |

Note: The average MS of OA factors are summarized by Hu (2012). The MS of OA factors resolved in studies over China are from He et al. (2010, 2011), Hu et al. (2013, 2016) and Huang et al. (2010, 2011). Other MS of OA factors are from AMS Spectral Database (Unit Mass Resolution). Specifically, the published spectra used for the average MA of each OA factor are listed as follows. **LV- and SV-OOA:** He et al., 2011; Hu et al., 2016; Huang et al., 2011. **HOA:** Aiken et al., 2009; He et al., 2011; Hu et al., 2016. **BBOA:** Aiken et al., 2009; He et al., 2010, 2011; Huang et al., 2011; Lanz et al., 2008; Ng et al., 2010; Weimer et al., 2008; **COA:** He et al., 2010; Hu et al., 2016; Huang et al., 2010; Mohr et al., 2009; **CCOA:** Hu et al., 2013; Vehicle-OA: Canagaratna et al., 2004; Mohr et al., 2009.

**Table S3** Correlation coefficients (Pearson's R) of OA factors with gaseous and aerosol species. Correlation coefficients higher than 0.60 are in bold.

[revised manuscript text omitted]

---

## Author Response (AR2)

**Point-to-point Response to the Comments of Editor**

The authors have worked hard to respond to and properly address Reviewer comments. We find several ways to improve the readability of the manuscript. The editor will review once more prior to technical/editorial corrections.

The authors have worked hard to respond to Reviewer comments. We find several ways to improve the manuscript. The editor will review once more prior to technical/editorial corrections. Comments below:

**Response:** We appreciate the Editor's comments. Following the Editor's suggestions, we have corrected the typos, checked through the manuscript carefully and revised the manuscript to make the expression clear.

Please see the following responses and the changes in the revised manuscript.

**Abstract:**

1. Page1, Line 10: This first sentence is worded awkwardly with pollution and pollutants appearing so close together. Since it is the first sentence, I suggest changing it to "Severe air pollution in Asia is often the consequence of a combination of large anthropogenic emissions and adverse synoptic conditions".

**Response:** Page 1, Line 14, the sentence was changed as the Editor suggested.

2. Page 1, Line 22: where the authors state "… secondary inorganic correlated well with relative humidity", please put the quantitative metrics that justify this (e.g., R2 and p values).

**Response:** Page 2, Line 5, **"(Pearson r=0.415−0.555, p<0.01)"** was added.

3. Page 1, Line 22: please change 'indicating" to "suggesting".

**Response:** Page 2, Line 5, changed.

4. Page 2, Line 1: The first sentence is awkward, I suggest rewording to:

"As the photochemical age of OA increased with higher oxidation state, secondary organic aerosol formation contributed more to OA." I encourage the authors to use the word 'significantly' in the sentence if they can demonstrate the change is statistically significant with a t-test or other appropriate statistical test that indicates significance.

**Response:** Page 2, Line 6, the sentence was changed, and the word "*significantly*" was removed.

**Introduction:**

5. Page 2, Line 18: the sentence that begins "The highest annual mean" … it's unclear what the authors are working to convey here and so I am not able to tell if it fits.

**Response:** Page 2, Line 23, "The highest annual mean aerosol optical depth (AOD) in the Sichuan Basin from 2000 to 2010 across China reflected the importance ..." was revised into "***The annual mean value of the aerosol optical depth (AOD) in the Sichuan Basin from 2000 to 2010 was the highest across China, reflecting the importance …***".

6. Page 2, Line 20, "and high RH" should be removed from the sentence.

**Response:** Page 3, Line 2, the words "*and high RH*" was kept because "high RH" is also one of the factors causing the low visibility in Chongqing.

7. Page 2, Line 20, the sentence beginning "Since the 2000s …" is unclear and needs revisions.

**Response:** Page 3, Line 2, "*Since the 2000s, the air quality has been aggravated ...*" was revised into "***The air quality has been aggravated …since the year 2000.***".

8. Page 3, Line 1: This 1st sentence should be reworded and I suggest the following: "Degraded air quality and variability of fine particle concentrations in the Sichuan Basin arises from substantial local anthropogenic emissions, regional photochemistry and are aggravated by topographical and meteorological conditions."

**Response:** Page 3, Line 6, the sentence was changed into "***Degraded air quality and variability of fine particle concentrations in the Sichuan Basin arise from substantial local anthropogenic emissions, regional photochemistry, and are aggravated by topographical and meteorological conditions***" according to the Editor's suggestion.

9. Page 3, Line 11: Biomass burning is already mentioned. The authors should remove the sentence here because it does not fit here. The authors can move the references up in the paragraph for the other biomass burning sentences.

**Response:** Page 3, Line 15, the sentence "*Further, biomass burning contributes importantly to air pollution in the Sichuan Basin (Wang et al., 2013; Yang et al., 2011)*" was removed. "*Secondary formation from biomass burning emissions can significantly influence local and regional air quality, atmospheric processes, and even climate change (Niu et al., 2016).*" was moved before "*Though several published papers ...*"

10. Page 3, Line 14: what makes the high time resolution measurements necessary I think it the episodic nature of emissions and meteorological conditions

**Response:** Page 3, Line 18, "*Therefore*" was changed into "***Due to the episodic nature of primary emissions and meteorological conditions***".

11. Page 3, Line 18: what does "concentration of OA in molecular level" mean?

**Response:** Page 3, Line 21, "*the concentration of OA in molecular level*" was revised into "***the concentration of OA quantified at the molecular level***".

12. Page 3, Line 20: it would help your argument to point out that levoglucosan is a tracer for biomass burning and local residential burning is the likely source of ambient observations. Further, by "occupied about", do the authors mean "contributed approximately 90% of identified sugar mass"?

**Response:** Page 3, Line 24, "*High levels of levoglucosan that were most likely emitted from residential biomass burning, occupied around 90% of total identified sugars (700 ng m$^{-3}$)*" was revised into "***High levels of levoglucosan, as a key tracer of biomass burning, that contributed approximately 90% of total identified sugars (700 ng m$^{-3}$), were most likely emitted from residential burning***".

13. Page 3, Line 21: instead of "drew that about" put "concluded that approximately".

**Response:** Page 4, Line 2, changed.

14. Page 3, Line 24, "ratio" should be "ratios"

**Response:** Page 4, Line 4, corrected.

15. Page 6, Line 13, Put "the" in front of 'airstream', remove "in", remove "subsequently" and move "isokinetically" to where subsequently was, remove the comma (",") after L min$^{-1}$.

**Response:** Page 6, Line 18, corrected.

16. Page 6, Lines 16/17. Black carbon is primary and emitted directly to the atmosphere. Secondary particles form in the atmosphere. Discussion of BC formation is therefore confusing and this sentence must be re-worded.

**Response:** The sentence is reworded from the original text "*BC particles are largely found in the Aitken mode (i.e., less than 100 nm diameter) and the accumulation mode because of their formation mechanism*" in Bond et al. (2013). It is used to explain that the mass closure of PM$_1$ measured by AMS and MAAP was applicable (also refer to response to **Comment 18**).

Page 6, Line 21, in the revision, "*formation*" was changed into "***generation***". In order to follow fluently, "*Atmospheric BC particles are mostly in the Aitken and accumulation modes (i.e., in the submicron range) because of their formation mechanisms (Bond et al., 2013)*" was revised into "***Because of the generation mechanism of atmospheric BC particles, they are mostly in the Aitken and accumulation modes, i.e., in the submicron range (Bond et al., 2013)***".

17. Page 7, Line 17: can the authors provide context for "…ratios of OA were biased low". Such a statement requires explanation and justification.

**Response:** Page 7, Line 21, "***because low H$_2$O$^+$/CO$_2$$^+$ ratios in OA were widely used (Chen et al., 2015)***" was added.

18. Page 8, Line 13: The authors give the mass concentrations of PM$_1$ (sum of AMS and MAAP), but in the methods they state the MAAP employed a PM$_{2.5}$ cutpoint cyclone. Can they resolve this discrepancy?

**Response:** Ambient BC particles are largely found in the Aitken and accumulation modes (i.e., in the

submicron range) because of their formation mechanism (Bond et al., 2013; Huang et al., 2012b; Rose et al., 2006). The sum of non-refractory species measured by HR-ToF-AMS and BC measured by instruments such as MAAP or aethalometer with the cut-size of 2.5 µm is often treated as total $PM_1$ in previous studies (Huang et al., 2010, 2012a, 2013; He et al., 2011; Hu et al., 2013, 2016). In this study, the morphology of individual particles also indicated that the sizes of soot particles were less than 1 µm (Fig. 4). So we thought this match has little influence on $PM_1$.

We addressed that "***Because of the generation mechanism of atmospheric BC particles, they are mostly in the Aitken and accumulation modes, i.e., in the submicron range (Bond et al., 2013)***" in Page 6, Line 21.

**Results and Discussion**

19. Page 9, line 8: in the results the authors state "local emissions may result from smoking bacon with biomass burning". This may be true, but it is odd for the authors to introduce a new topic and idea near the end of the paper.

**Response:** The sentence "*The local emissions may result from smoking bacon with biomass burning, a traditional and common method of preserving pork and sausages in the Sichuan Basin in winter.*" was removed here. "***Making smoked bacon with biomass (firewood) burning, a traditional and common method of preserving pork in the Sichuan Basin in winter***" was added in Page 13, Line 5.

20. Page 9, Line 10: step-wise is a good descriptor, step-wisely is not really a word. This sentence needs to be re-worked.

**Response:** Page 9, Line 12, "*step-wisely increased*" was changed into "***increased stepwise***".

21. Page 9, line 24: I do agree with the authors that water uptake and subsequent condensed/aqueous phase chemistry could increase inorganic mass concentrations however, I do not think hygroscopic growth (i.e., water uptake) alone would increase SNA mass concentrations. Water mass would most likely be removed during measurement, and if not, would not be counted in the specific chemical categories. The authors need to reword this more carefully.

**Response:** The sentence "*the humid air after the precipitation may favor the aqueous-phase secondary formation and hygroscopic growth of SNA, causing...*" was changed into "***the humid air after the precipitation may favor the hygroscopic growth of SNA and subsequent condensed/aqueous phase chemistry, causing…***".

22. Page 10, Line 17, instead of 'around 600-800', please rephrase to something like "peaks centered at 600-800". Can the authors provide justification (e.g., reference) for why this mode implies internally mixed particles?

**Response:** Page 10, Line 19, the phrase "*around 600−800 nm*" was revised into "***centered at 600−800 nm***". The similar size distribution shapes of organics and secondary inorganics rather than the mode

implied the internally mixed conditions of particles. "*(Hu et al. 2013, 2016)*" was added as references.

23. Page 10, Line 18: Can the authors provide context, explanation and justification (e.g., with references) for why "This was consistent with the morphology and mixing state of single particles, mostly spherical and in internal mixing state…" The statement does not follow exactly.

**Response:** This statement is a description of the morphology and mixing state of individual particles under the transmission electron microscope, to confirm that the airborne particles were in internal mixing state.

Page 10, Line 19, the sentence "*This was consistent with the morphology and mixing state of single particles, mostly spherical and in internal mixing state*" was changed into "***The TEM images (Fig. 4a-d) also demonstrated that the individual particles were mostly spherical and in the internal mixing state***".

24. Figure 1. The blue wording is hard to read and I fear it will be illegible upon publication. I do like locating the cities in the context of the emission inventory. Perhaps arrows pointing the location so the words are outside the red/black areas.

**Response:** Fig. 1 was revised as follows.

[Figure]

*Figure 1. Location of the observation site in Ziyang in the Sichuan Basin. Back-trajectories of air masses at the site calculated by HYSPLIT model are illustrated as lines (circles marking 24-h intervals). The map of China is color-coded according to residential OC emissions in January 2010 modeled by Multi-resolution Emission Inventory for China (MEIC, http://www.meicmodel.org).*

25. Figure 2. The x-axis legend is very difficult to read. Perhaps writing 'December 2012' and 'January 2013' below and only having numbers for the days on the axis would make things more clear.

**Response:** Fig. 2 and Fig. 6 were revised as follows according to the Editor's suggestion.

[Figure]

*Figure 2. Time series of meteorological parameters and concentrations of chemical compositions in submicron aerosols during the campaign. (a) Wind speed (WS), relative values; (b) relative humidity (RH), temperature and atmospheric pressure; (c) concentrations of chemical compositions in submicron aerosols. Short-term precipitation events are marked by the light blue arrows.*

[Figure]

*Figure 6. Time series of OA fractions and external tracers (sulfate, nitrate, BC, chloride and acetonitrile).*

[revised manuscript text omitted]